# Seismic Effect of Marine Corrosion and CFRP Reinforcement on Wind Turbine Tower

**Daoyong Wang** [1,2] **, Bo Song** [1,2,*]**, Shuo Diao** [1,3]**, Chao Wang** [1,2] **and Chenhu Lu** [1,2]

1    School of Civil and Resource Engineering, University of Science and Technology Beijing, Beijing 100083, China
2    Beijing International Cooperation Base for Science, Technology-Aseismic Research of the Rail Transit Engineering in the Strong Motion Area, Beijing 100083, China
3    CABR Testing Center Co., Ltd., Beijing 100083, China
*    Correspondence: songbo@ces.ustb.edu.cn; Tel.: +86-1336-6236-292

**Featured Application: Global warming makes many countries pay more attention to the use of wind energy. With the development of offshore wind power generation, the offshore wind turbine tower is facing the dual effects of marine corrosion and earthquake. The purpose of this paper is to study the seismic performance of wind turbine tower after corrosion and the effect of CFRP (Carbon Fiber Reinforced Polymer) reinforcement. It provides a reference for the development of offshore wind turbine towers.**

**Abstract:** The offshore wind turbine tower, which has been in the marine corrosive environment for a long time, often buckles and collapses under the earthquake records. In order to study the influence of marine corrosion and CFRP reinforcement on the seismic performance of wind turbine tower structures, the horizontal displacement, horizontal acceleration and acceleration change rate of wind turbine towers are studied through numerical simulation and shaking table tests. The results show that the influence of earthquake type on the dynamic response of the wind turbine tower is different. The response values of acceleration and displacement under far-field earthquakes are larger than those of other earthquake types. The increase in PGA has a greater impact on the structural response range in the near-field earthquake. Corrosion defects not only increase the sensitivity of the wind turbine tower structure to seismic response but also have different effects on the location and development of structural plastic hinges. For the structure without corrosion defects, the plastic hinge appears at the connection between the tower and the foundation, while in the corrosion structure, the plastic hinge appears in the corrosion area. Corrosion defects increase the nonlinear development of structures, especially under far-field earthquakes. CFRP reinforcement can effectively reduce the displacement effect of the top of the structure and enhance the seismic performance of the corroded wind turbine tower.

**Keywords:** wind turbine tower; corrosion defects; CFRP reinforcement; dynamic response; shaking table test

## 1. Introduction

With the global warming caused by the greenhouse effect becoming more and more serious, the development of renewable energy is becoming more and more important. In renewable energy technology, in order to meet the growing demand for electricity, the use of wind turbines is growing exponentially. However, with the increase in generator design and power generation capacity, various engineering problems are gradually exposed [1–3]. With the rapid development of wind power generation, the number of wind power towers is increasing year by year, but the development of the research on the seismic performance of wind power towers is relatively lacking, especially for wind turbine towers suffering from marine corrosion, the research on the seismic performance after corrosion needs to be carried out urgently [4–6]. After the 2003 edition of "guidelines for structural design

of wind power equipment supports" in Japan, considering the large-scale development trend of wind turbine towers, the code was revised in 2010 [7]. The main revised content includes: for wind turbine towers with a height of over 60 m, the structural calculation of the tower structure, anchorage position and foundation should be able to resist extremely rare earthquakes. The performance design method based on the limit state design method is adopted in the design. Therefore, it is necessary to study the influence of different corrosion degrees and earthquake types on the seismic performance and collapse of offshore wind turbine towers under rare earthquakes.

Katsanos et al. [8] comprehensively expounded the correlation of wind turbine seismic disasters, studied the key factors of wind turbine seismic design, and pointed out that more experimental studies should be carried out to clarify the seismic performance of wind turbine towers. Yazhou Xu et al. [9] studied the influence of near-field pulse type earthquakes and welding defects on wind turbine towers with 1.5 MW onshore wind turbine towers. The results show that welding defects can significantly reduce the seismic intensity required for the nonlinear response of wind turbine towers, especially under pulse type near-field earthquake excitation, the structural nonlinearity with welding defects develops faster and higher. Atul Patil et al. [10] studied the buckling and overturning characteristics of wind turbine towers under the action of pulsed near-fault earthquakes and far-fault earthquakes. It is found that the wind turbine tower is the most prone to overturning under earthquakes, and the yield of tower is the second possible failure mechanism, followed by the development of permanent deformation and overall buckling of tower. Qianqian Ren et al. [11] emphasized the importance of incorporating the pulsed near-field effect into the seismic design of wind turbines, especially for wind turbines located near active faults. Shuai Huang [12] proposed a simplified calculation model for the dynamic interaction of water, sea ice and wind turbine towers under seismic action, which avoids the solution of complex nonlinear equations and reduces the calculation burden. According to the research on the corrosion characteristics of steel structures in seawater, due to the alternation of dry and wet, the splash zone has the fastest corrosion rate and the most serious corrosion degree [13–17]. Masayuki Tai et al. [18] used FEM (Finite Element Method) to study the seismic performance and buckling characteristics of corroded steel pipe piles. The analysis results show that the damage location of steel pipe piles has little effect on the bearing capacity. Khodabux et al. [19] studied the corrosion rate of steel structures in Western wind farms, and the highest corrosion rate of steel was 0.83 mm/year.

As mentioned above, there is little research on the corrosion of offshore wind turbines tower at present. Most of the existing research focuses on different earthquake types, but the influence of corrosion defects on the seismic performance of the structure is not considered at the same time. Therefore, based on the consideration of corrosion defects, this paper studies the influence of near-field and far-field earthquakes on the dynamic response and collapse of the wind turbine tower structure. At the same time, CFRP is applied to the reinforcement of corroded wind turbines, and the effect of CFRP reinforcement is verified.

## 2. Establishment and Verification of FE Model

A Hua Rui SL3000-3WM wind turbine tower was selected as the research object. The structural dimensions and geological conditions are shown in Figure 1. The height of the tower is 75 m, and the buried depth of the foundation is 34 m. The diameter varies from 4.05 m to 3.07 m along the tower height. The maximum thickness of the bottom is 50 mm, and the minimum thickness of the top is 20 mm. In order to study the influence of different earthquakes and corrosion degrees, the FE model of the wind turbine tower is established and verified by comparison with field test. The seismic dynamic response and collapse analysis of wind turbine structures are calculated by the FE (Finite Element) model.

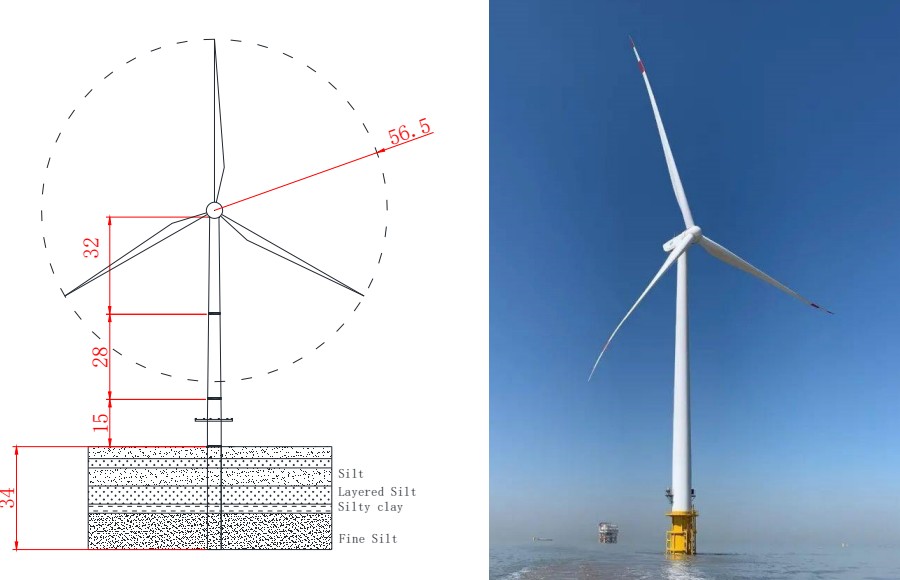

**Figure 1.** Structural dimensions and geological conditions of wind turbine tower.

### 2.1. Establishment of FE Model of Wind Turbine Tower

The methods of simulating soil structure interaction (SSI) include the fictitious anchor point method, the m method, the P-y curve method and the actual soil modeling method. Considering the large amount of calculation of seismic time history analysis, the actual soil modeling method is not applicable to the analysis of this study. In addition, the m method and the P-y curve method need to add many nonlinear springs to the structure, which will submerge the influence of corrosion and CFRP reinforcement on the structure in the deformation of nonlinear springs. In practical engineering, the pile foundation of wind turbine towers is usually embedded in the foundation or rock stratum. Therefore, the assumed embedded point method is used to consider the soil structure interaction (SSI). As shown in Figure 2, calculate the embedded depth of long elastic pile according to the format (1)~(2) Code for pile foundation of harbor engineering [20].

$$t = \eta T \tag{1}$$

$$T = \sqrt[5]{\frac{E_p I_p}{m b_0}} \tag{2}$$

Here: $\eta$ is the coefficient, taking 1.8–2.2, the smaller value is taken when the pile top is hinged or the free length of the pile is larger, and the larger value is taken when the pile top angle does not rotate or the free length of the pile is smaller; $T$ is the relative stiffness of the pile; $E_p$ is the elastic modulus of the pile material; $I_P$ is the moment of inertia of the pile section; $m$ is the proportional coefficient of the horizontal resistance coefficient of the foundation soil at the pile side increasing with the depth; $b_0$ is the converted width of the pile.

The FE model of the wind turbine tower without corrosion is established by ABAQUS. The 8-node solid element C3D8R is selected for the tower structure, the material is Q345B steel. The mechanical parameters of steel are shown in Table 1. In order to study the mechanical behavior of the structure under earthquakes, the strengthening constitutive model of steel under cyclic action has been adopted [21,22].

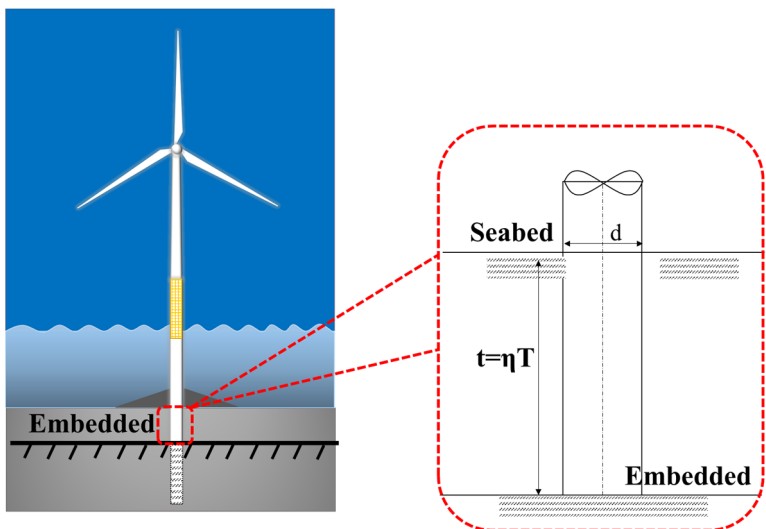

**Figure 2.** Hypothetical embedded point method.

**Table 1.** Parameters of the constitutive law of steel.

| Properties | Values | Properties | Values |
|---|---|---|---|
| Density (kg/m$^3$) | 7850 | $C_{kin \cdot 1}$ | 7993 |
| Elastic modulus (MPa) | $2.06 \times 10^5$ | $C_{kin \cdot 2}$ | 6773 |
| Poisson's ratio | 0.28 | $C_{kin \cdot 3}$ | 2854 |
| Yield strength (MPa) | 345 | $\gamma_1$ | 175 |
| Yield surface Equivalent stress (MPa) | 429 | $\gamma_2$ | 116 |
| Hardening parameter | 1.2 | $\gamma_3$ | 34 |

Note: $C_{kin\,i}$ and $\gamma_i$ are the material parameters of the hybrid follow-up strengthening constitutive model. $C_{kin\,i}$ is the strain-hardening modulus, which is the plastic modulus at the beginning of plastic deformation, and $\gamma_i$ controls the rate at which the strain-hardening modulus decreases with the plastic strain.

Ke S.T. et al. [23,24] took the 5 wm wind turbine tower as the research object and studied the coupling effect of blade and tower tubes on the dynamic response of the wind turbine tower structure. It is found that the influence of the blade on the dynamic response of the wind turbine structure cannot be ignored. Therefore, in order to ensure the accuracy of calculations, the blade structure is established in ABAQUS, and the material is simplified according to the equal mass. The equivalent density is 171.58 kg/m$^3$. According to the principle of equal mass, the density of the engine room and the wheel hub are equivalent. The weight of the cabin is 131.43 tons, and the hub is 27.78 tons. The pile bottom is completely fixed. Tie constraints are used between the blade and the hub and in the engine room. Because the grid generation will affect the calculation speed and accuracy, on the premise of ensuring the calculation accuracy, the tower structure is encrypted, and other parts of the grid are simplified. When establishing the finite element model, refer to the research of Koulatsou, K.G. et al. and Mohammad-Amin Asareh et al. [25,26]. The bolted connection is simplified as a steel ring inside the tower to reduce the number of elements in the finite element model. The overall finite element model of the structure is shown in Figure 3.

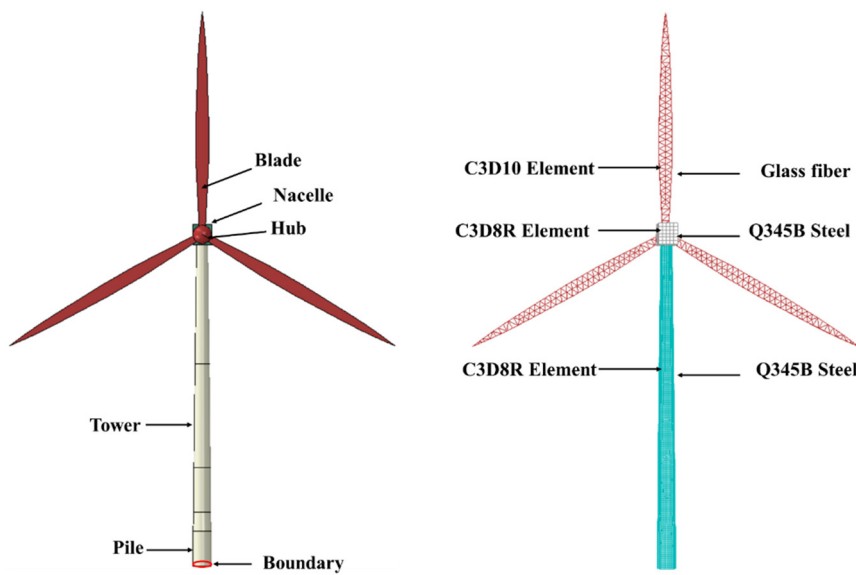

**Figure 3.** FE model of wind turbine tower structure.

### 2.2. FE Model with Marine Corrosion

The corrosion area of steel plate and steel pipe pile exposed to the marine environment is usually divided into the marine atmospheric zone, spray splash zone, ocean tidal zone, seawater immersion zone and submarine mud zone [27]. Paik et al. further calibrated the parameters of the Weibull corrosion model with the measured data from a ship with a 10-year service period [28]. Bai et al verified the applicability of the model in offshore platforms and used the model to analyze the corrosion effects of different areas on jacket platforms [29]. In this study, according to the corrosion probability model of the Weibull function proposed by Yang Hongqi [30] and Qin Shengping [31], the corrosion thickness of the tower structure in the droplet zone is calculated. The expression of the model is as follows:

$$d_n(t) = \begin{cases} 0; & 0 < t < T_{st} \\ d_\infty \left\{ 1 - \exp\left[-((t - T_{st})/\eta)^\beta\right] \right\}, & T_{st} < t < T_L \end{cases} \tag{3}$$

Here: $d_n(t)$ is corrosion depth corresponding to time; $d_\infty$ is Corrosion limit depth; $t$ is corrosion duration; $T_{st}$ is the beginning of corrosion; $T_L$ is Structure life or structure maintenance time; $\eta$ and $\beta$ is Environmental parameters of marine corrosion.

Write the Python program according to the execution logic shown in Figure 4. After inputting the initial modeling parameters, the program determines the diameter distribution of corrosion pits according to the normal distribution law (Formula (3)). Calculate the number of corrosion pits and the loss rate of corrosion volume. When the number of generated corrosion pits reaches the target corrosion rate, the generation of corrosion pits is stopped. Otherwise, the program adjusts the number of corrosion pits until the target corrosion rate is met. After the number of corrosion pits is determined, the random function is used to generate the location of each corrosion pit, when two corrosion pits overlap, Z and $\theta$ Value are computed until all corrosion pits are randomly distributed to the whole structure without coincidence. Figure 5 shows the FE model with random corrosion pits established by Python and ABAQUS.

$$f(h) = \frac{1}{\sqrt{2\pi}\sigma} e^{-\frac{(h-\mu)^2}{2\sigma^2}} \tag{4}$$

Here: $\sigma$ is Standard deviation; $\mu$ is average value; $h$ is corrosion pit depth.

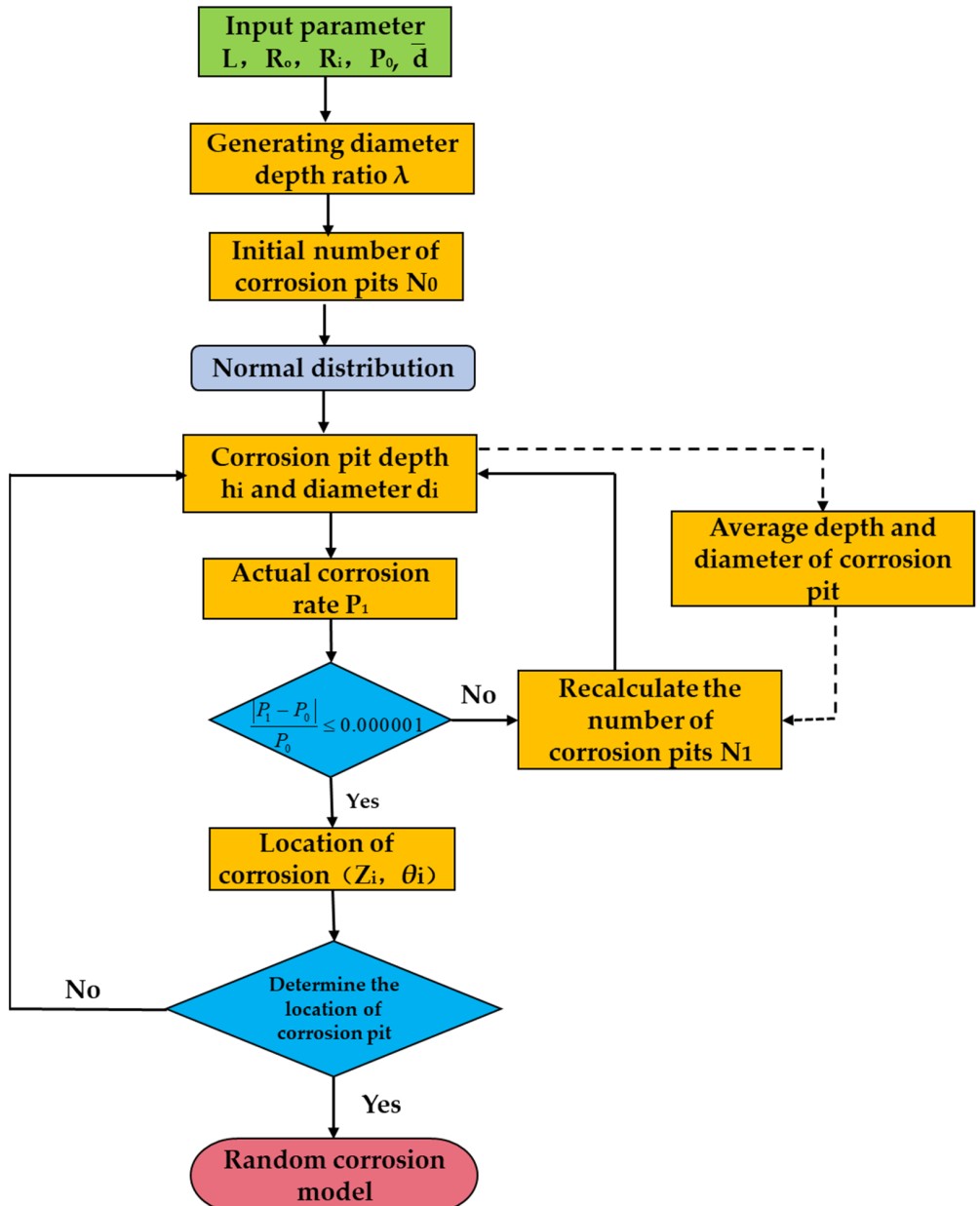

**Figure 4.** Python program execution process.

The finite element model with corrosion for 10 years, 20 years and 30 years is established according to the above method. The wall thickness of the tower varies with time, as shown in Figure 6.

The CFRP model is created using the S4R unit in the ABAQUS software. Use Create Composite Layup to set the number of fiber layers, the thickness of each fiber layer, and the ply angle of each fiber layer. The material properties of CFRP are expressed by the "ENGINEERING CONSTANTS" model and the "HASHIN DAMAGE" model.

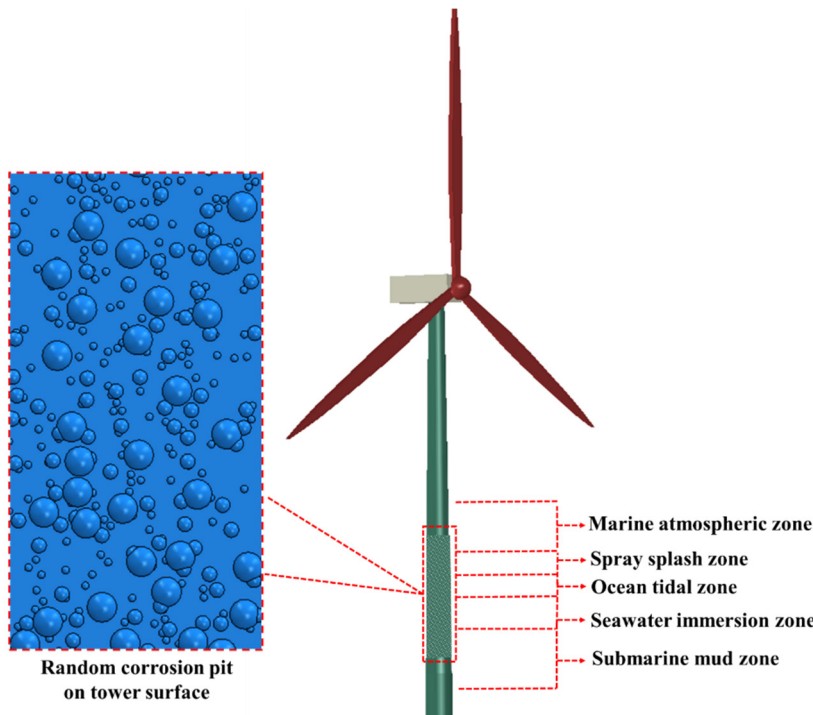

**Figure 5.** FE model with random corrosion.

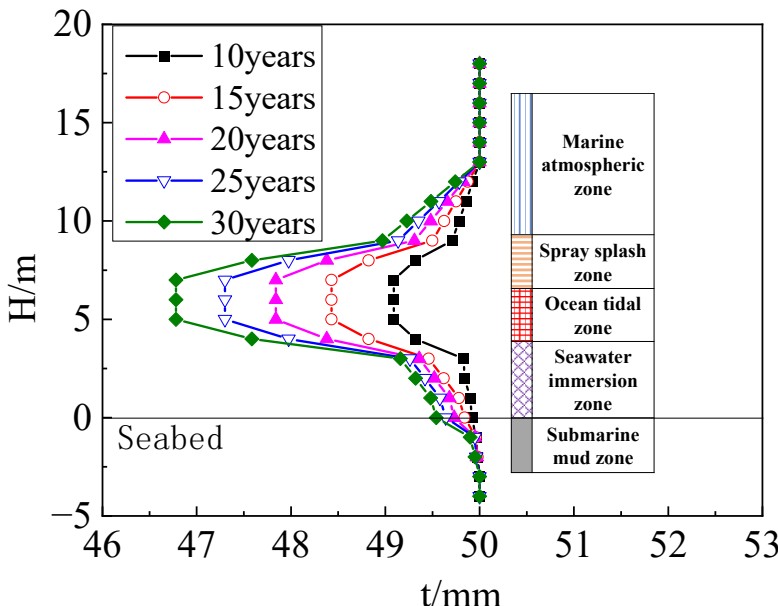

**Figure 6.** The wall thickness of the tower varies with time.

### 2.3. Dynamic Characteristics and Model Verification of Wind Turbine Tower

Based on the Lanczos method of ABAQUS, the mode shapes and natural frequencies of wind turbine tower structures are analyzed. The first four orders of mode vibration are shown in Figure 7. The first mode of vibration of the structure is left and right oscillation of the blade and the superstructure, the second mode is the front and rear vibration, the third mode is mainly the torsional deformation of the blade and the superstructure, and the fourth mode is the bending deformation of the structure.

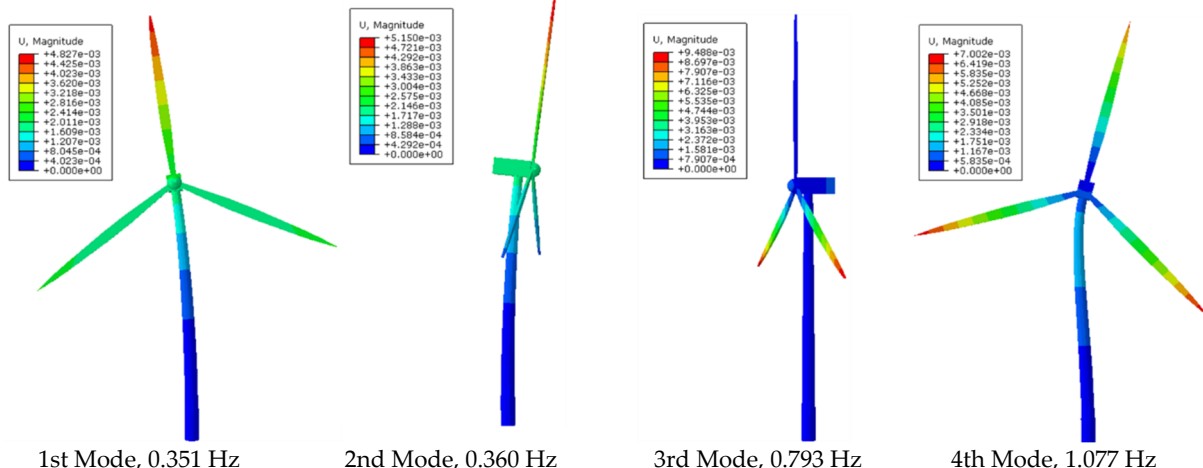

| 1st Mode, 0.351 Hz | 2nd Mode, 0.360 Hz | 3rd Mode, 0.793 Hz | 4th Mode, 1.077 Hz |

**Figure 7.** Model shapes of the tower.

To verify the correctness of the finite element model, the calculated natural frequency of the FE model is compared with the structural natural frequency data measured by the "braking test" of the wind turbine. Two wind turbine towers are selected as experimental objects, namely 6# and 15#. The "braking test" refers to the emergency shutdown of the running fan system. The excitation effect is generated by the emergency shutdown, so as to measure the vibration characteristics of the whole wind turbine tower. The excitation effect is generated by the emergency shutdown to measure the vibration characteristics of the entire wind turbine tower. The statistics of identified natural frequencies and damping ratios of the structure obtained through the "braking test" are shown in Table 2.

**Table 2.** Statistics of identified natural frequencies and damping ratios.

| Test | 6# Wind Turbine Tower | | | 15# Wind Turbine Tower | | |
|---|---|---|---|---|---|---|
| | Frequency | Natural Period | Damping Ratio (%) | Frequency | Natural Period | Damping Ratio (%) |
| 1 | 0.319 | 3.135 | 1.01 | 0.331 | 3.021 | 0.99 |
| 2 | 0.318 | 3.145 | 1.02 | 0.327 | 3.058 | 1.00 |

Table 3 shows the comparison of natural frequencies of wind turbine. The first-order natural frequency of the structure measured in two "braking tests" of 6# wind turbine is 0.319 Hz and 0.318 Hz, respectively. The difference between the calculated value of the finite element model and the measured value is 9.1% and 9.4%, respectively. The first natural frequency of the structure, measured twice by 15# wind turbine, is 0.327 Hz and 0.331 Hz, respectively. The difference between the calculated value of the finite element model and the measured value is 5.7% and 6.8%, respectively, both within the allowable range of the project, which can prove the rationality of the finite element model.

**Table 3.** Comparison of natural frequencies of wind turbine.

| Wind Turbine Tower | Test | FE Value (Hz) | Measured Value (Hz) | Difference Value (%) |
|---|---|---|---|---|
| 6# | 1 | 0.351 | 0.319 | 9.1 |
| | 2 | 0.351 | 0.318 | 9.4 |
| 15# | 1 | 0.351 | 0.331 | 5.7 |
| | 2 | 0.351 | 0.327 | 6.8 |

### 3. Numerical Analysis of Seismic Dynamic Response

*3.1. Selection of Earthquake Records*

According to the recommended method of the National Earthquake Hazards Reduction Program (NEHRP) [32] and the GB50011-2010 Code for seismic design of buildings [33], three different types of earthquakes are selected by using the Pacific Earthquake Engineering Research Center (Peer) seismic database [34]. They include the El Centro earthquake, the Nihonkai-Chubu earthquake and the Hyogoken-Nanbu earthquake. The Nihonkai-Chubu earthquake belongs to the long-term and long-term earthquakes, and the Hyogoken-Nanbu earthquake belongs to the near-field direct and lower type earthquakes. The characteristics of the three types of seismic records are shown in Table 4.

**Table 4.** The characteristics of different earthquake records.

| Earthquake | Station Location | Fault Distance (km) | PGA (cm/s$^2$) | PGV/PGA (s) | ML |
|---|---|---|---|---|---|
| El Centro | El Centro#9 | 6.09 | 341.7 | 0.10 | 6.9 |
| Nihonkai-Chubu | Akita meteorological station | 14.0 | 58.6 | - | 7.7 |
| Hyogoken-Nanbu | Takarazuka | 16.1 | 818.0 | 0.12 | 7.2 |

According to the provisions of the GB50011-2010 Code for the seismic design of buildings, the selected seismic records are adjusted to different PGA values (0.07 g, 0.2 g and 0.40 g), corresponding to different seismic intensities. The PGA values of 0.07 g, 0.2 g and 0.40 g respectively represent the small earthquake, design foundation earthquake and large earthquake in the earthquake area with a magnitude of 8. For the El Centro and Nihonkai-Chubu earthquakes, the horizontal input is considered for seismic excitation, while for the Hyogoken-Nanbu earthquake, the horizontal and vertical bidirectional input is considered for seismic excitation. According to the code for seismic design of buildings, the peak value of vertical seismic acceleration is 0.65 times the horizontal peak value. Taking the model without defects as an example, the influences of different near-field and far-field earthquakes on the dynamic response of structures are studied and compared.

*3.2. Influence of Earthquake Types and PGA*

The displacement time history curves of the top of the wind turbine tower structure under different earthquake actions are extracted, as shown in Figure 8. It can be clearly seen from Figure 8 that under the action of the El Centro earthquake, when PGA = 0.07g, the maximum displacement at the top of the wind turbine tower appears at 12.79 s, the maximum value is 0.051 m; when PGA = 0.2 g, the maximum displacement at the top of the wind turbine tower appears at 6.02 s, the maximum value is 0.122 m; when PGA = 0.4 g, the maximum displacement at the top of the wind turbine tower appears at 6.02 s, the maximum value is 0.245 m.

Under the action of Nihonkai-Chubu earthquake, the time history curve of tower top is shown in Figure 9. When PGA = 0.07 g, the maximum displacement of the top of the wind turbine tower appears in 41.55 s, the maximum value is 0.211 m; when PGA = 0.2 g, the maximum displacement of the top of the wind turbine tower appears in 41.59 s, the maximum value is 0.493 m; when PGA = 0.4 g, the maximum displacement of the top of the wind turbine tower appears in 41.56 s, the maximum value is 0.989 m.

Under the action of the Hyogoken-Nanbu earthquake, the time history curve of tower top is shown in Figure 10. When PGA = 0.07 g, the maximum displacement of the top of the wind turbine tower appears at 11.04 s, the maximum value is 0.031 m; when PGA = 0.2 g, the maximum displacement of the top of the wind turbine tower appears at 11.04 s, the maximum value is 0.085 m; when PGA = 0.4 g, the maximum displacement of the top of the wind turbine tower appears at 11.03 s, the maximum value is 0.170 m.

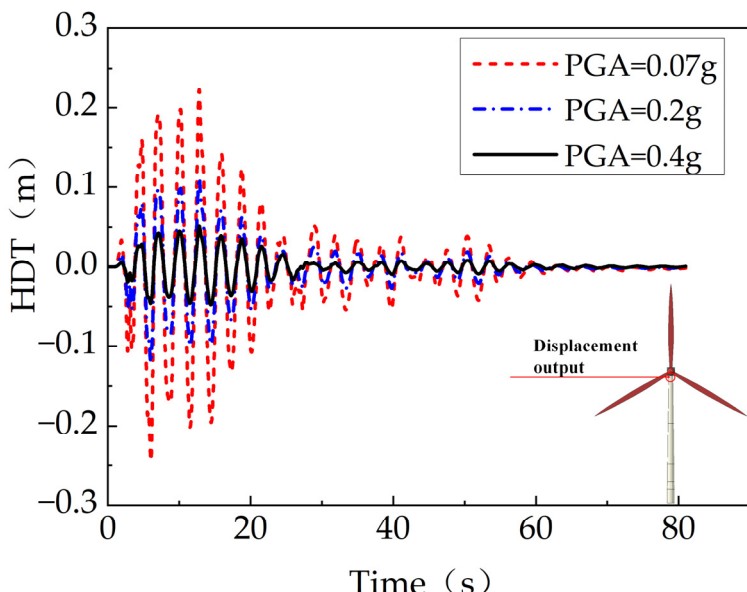

**Figure 8.** El Centro earthquake.

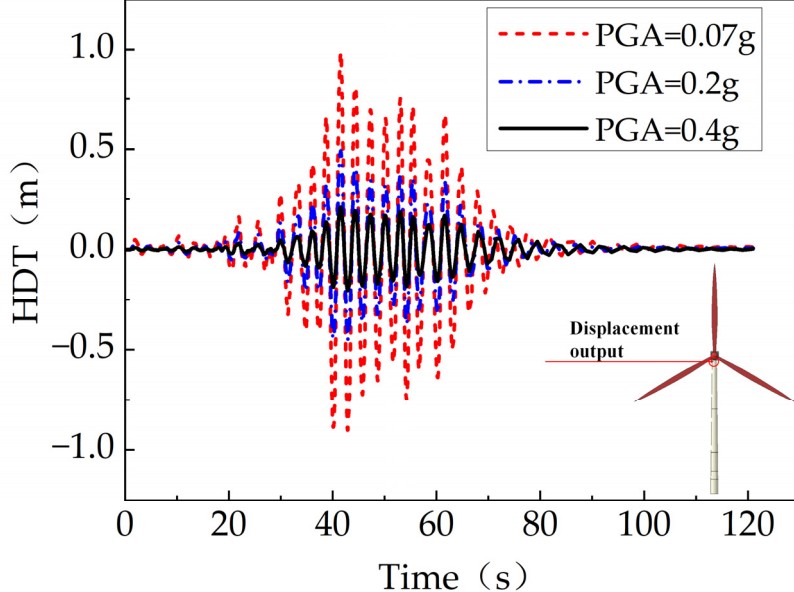

**Figure 9.** Nihonkai-Chubu earthquake.

Comparing the above results, it can be seen that the time of maximum displacement of the structure under the El Centro earthquake has changed, but the maximum displacement of the structure under the other two types of earthquakes has not changed, as shown in Figure 11a. In addition, the maximum displacement of the structure increases significantly under the far-field earthquake, which shows that the displacement caused by the far-field earthquake is more obvious for the structure with a longer natural vibration period. When PGA increases from 0.07 g to 0.2 gal, the maximum displacement of the El Centro earthquake and the Nihonkai-Chubu earthquake increases by 136% and 135%, respectively, and the maximum displacement of the tower increases by 174%, respectively, under the Hyogoken-Nanbu earthquake. Similarly, when PGA increases from 0.2 g to 0.4 g, the maximum displacement of the tower increases by 136% and 135%, respectively. The maximum displacement of the structure under the Hyogoken-Nanbu earthquake is 448% higher than that under the 0.07 g earthquake, which is significantly larger than that under the other two kinds of earthquakes. The structural dynamic response changes under

different earthquake records are shown in Table 5. Therefore, it can be seen that the displacement response of the structure under the near-field earthquake is larger. The peak acceleration at the top of the structure under three kinds of earthquake action can be seen from Figure 11b. Under the action of the Nihonkai-Chubu earthquake, the structural acceleration is obviously larger than that of the other two kinds of earthquake. However, the increment of peak acceleration with PGA under the El Centro earthquake and the Hyogoken-Nanbu earthquake is larger than the peak acceleration under the Nihonkai-Chubu earthquake.

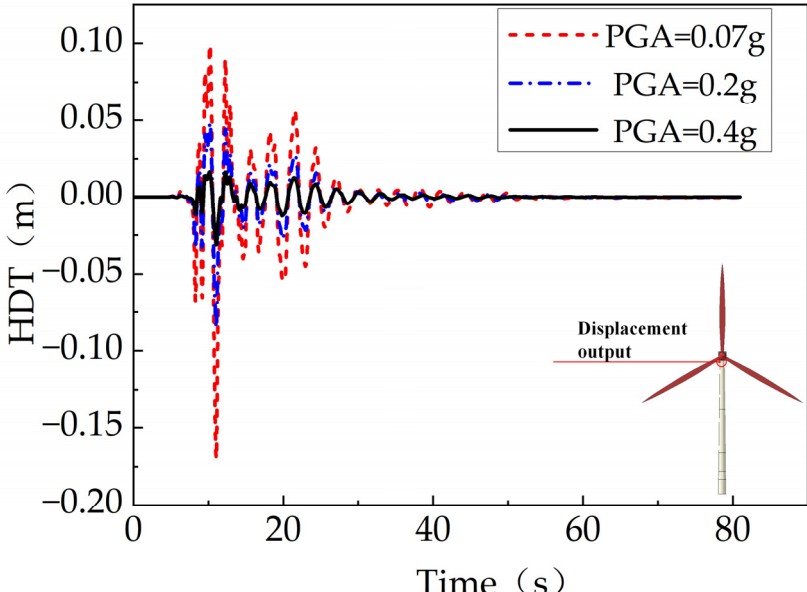

**Figure 10.** Hyogoken-Nanbu earthquake.

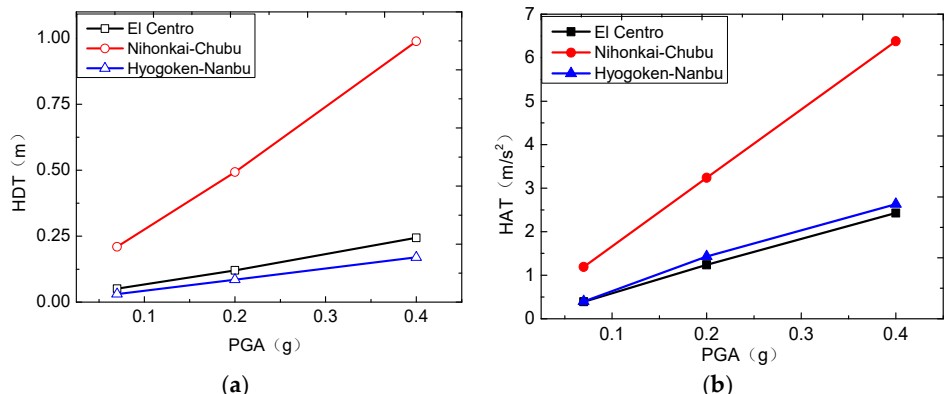

**Figure 11.** Comparison of dynamic response characteristics of wind turbine tower: (**a**) the maximum displacement; (**b**) the peak acceleration.

The above results show that the influence of near-field earthquakes and far-field earthquakes on the tall thin-walled structure of wind turbine tower is quite different. The acceleration and displacement responses are larger under a far-field earthquake, but the increase of PGA in a near-field earthquake has more influence on the variation range of structural response.

As shown in Figure 12: The dominant frequency of the El Centro seismic wave is 1.12 Hz, and the main energy of the Nihonkai-Chubu seismic wave is mainly concentrated in the range of 0~18 Hz, with a dominant frequency of 0.42 Hz. The main energy of the Hyogoken-Nanbu earthquake's seismic wave is mainly concentrated in the range of 0~15 Hz, and its predominant frequency is 1.36 Hz. It can be seen from the analysis that there are many high-frequency components of conventional earthquakes and near-site

motion, and the predominant frequency of the seismic wave is greater than 1 Hz. There are many low-frequency components of long-period earthquakes, the predominant period is less than 1 Hz, and it is close to the natural frequency of wind power towers. Therefore, the response of wind power tower structures to far-field earthquakes is more obvious.

**Table 5.** Dynamic response of structures under different seismic records.

| Case Condition | PGA | HDT (m) | Growth Rate | HAT (m/s$^2$) | Growth Rate |
|---|---|---|---|---|---|
| El Centro | 0.07 g | 0.051 | - | 0.388 | - |
| | 0.2 g | 0.120 | 136% | 1.240 | 220% |
| | 0.4 g | 0.244 | 378% | 2.430 | 526% |
| Nihonkai-Chubu | 0.07 g | 0.210 | - | 1.190 | - |
| | 0.2 g | 0.493 | 135% | 3.240 | 172% |
| | 0.4 g | 0.988 | 370% | 6.380 | 436% |
| Hyogoken-Nanbu | 0.07 g | 0.031 | - | 0.398 | - |
| | 0.2 g | 0.085 | 174% | 1.432 | 260% |
| | 0.4 g | 0.170 | 448% | 2.635 | 562% |

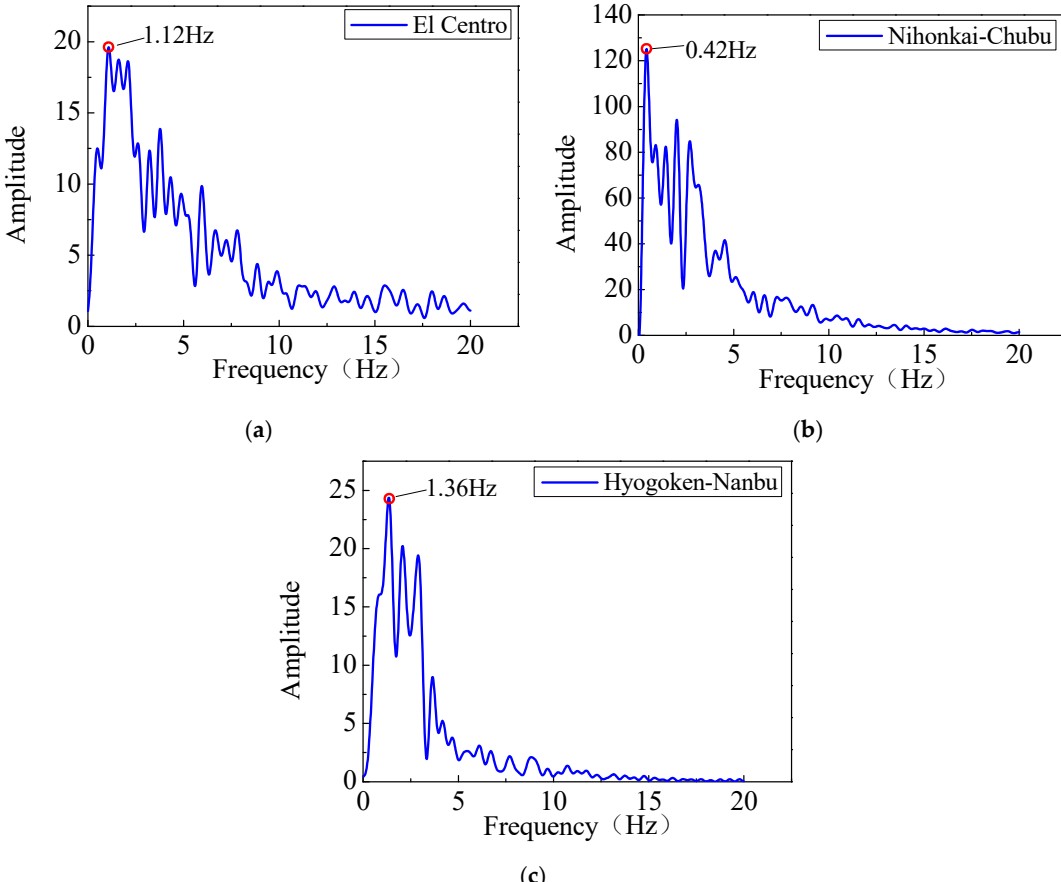

**Figure 12.** Fourier spectrum of seismic wave: (**a**) the El Centro earthquake; (**b**) the Nihonkai-Chubu earthquake; (**c**) the Hyogoken-Nanbu earthquake.

*3.3. Affect of Corrosion on Dynamic Response of Wind Turbine Tower*

The change rate of peak acceleration at the top of wind turbine tower structure is defined as APRC:

$$\text{APRC} = \frac{A_{TP} - A_{0P}}{A_{0P}} \tag{5}$$

Here: $A_{TP}$ is the peak acceleration of the top of the structure under the seismic action of peak acceleration P for the structure corroded for T years; $A_{0P}$ is the peak acceleration of the top of the structure without corrosion under the seismic action of the peak acceleration P.

Under the El Center earthquake, the peak acceleration at the top of the structure decreases with the increase in the service time of the structure. The peak acceleration of the top structure acceleration decreases more quickly after 10 years of service. With the increase of PGA, the peak acceleration of top structure increases first and then decreases. When PGA = 0.8~1.0 g, the top acceleration of the structure changes the most and then decreases gradually. Based on the analysis of Figure 13, it can be concluded that when the peak seismic acceleration reaches 1.0 g, the material stress at the top of the wind tower reaches 430 MPa, reaching the material yield state. It can be seen that the structural APRC gradually increases with the increase of PGA before the material at the top of the tower barrel yields. When the material at the top of the tower yields, the tower barrel enters the plastic state in some areas, and the structural APRC value decreases gradually.

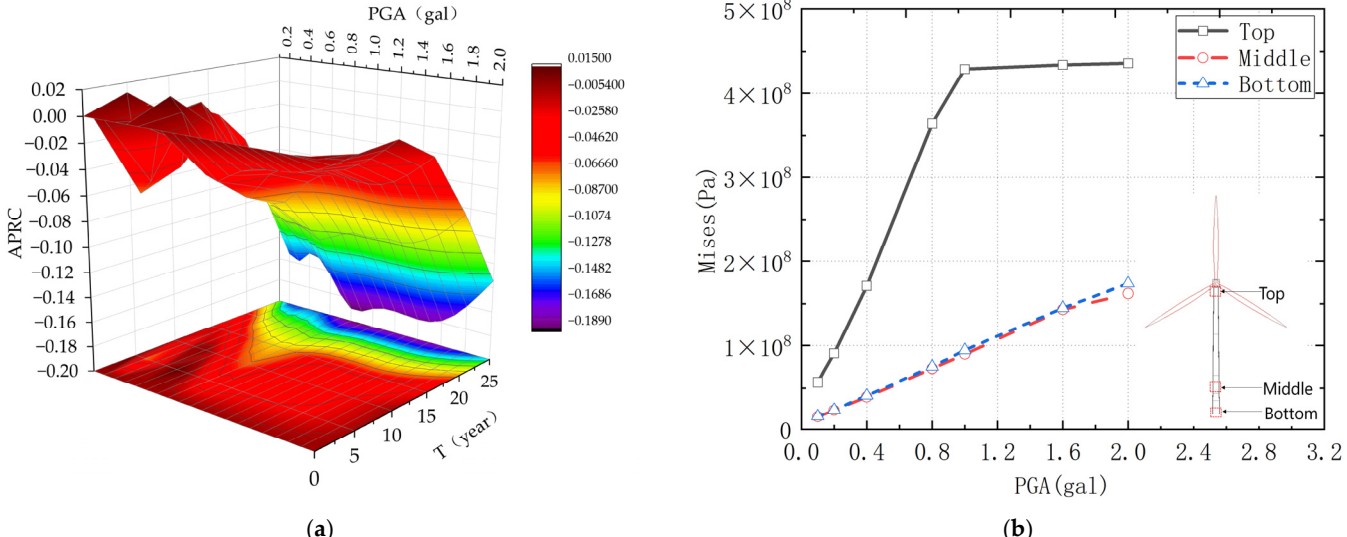

**Figure 13.** The El Centro earthquake: (**a**) structural APRC value; (**b**) maximum Mises stress.

It can be seen from Figure 14 that under the Nihonkai-Chubu earthquake, the change value of structural acceleration gradually increases with the increase of PGA when PGA is less than 0.6 g. When PGA is greater than 0.6 g, the change value of structural acceleration suddenly decreases. When PGA is greater than 0.8 g, the APRC value increases gradually with the PGA value again, and decreases again when PGA is greater than 1.2 g. Figure 14b shows that when PGA exceeds 0.6 g, the material at the top of the structure yields, the material at the top of the tower barrel yields between 0.8 and 1.2 g, and the structural stress at the corroded area and bottom of the tower barrel gradually increases and approaches the material yield stress. When PGA exceeds 1.2 g, the corroded area and bottom of the structure also reach the material yield. This phenomenon further proves that the structural dynamic response changes before and after material yields during earthquakes.

It can be seen from Figure 15 that under the Hyogoken-Nanbu earthquake, when the PGA is greater than 0.6 g, the APRC value increases. Compared with the El Centro earthquake, the peak acceleration of 0.8 g begins to yield because the Hyogoken-Nanbu earthquake contains the action of vertical acceleration. Since the material yield only occurs at the top of the tower, the APRC value of the structure only turns at 0.6 g.

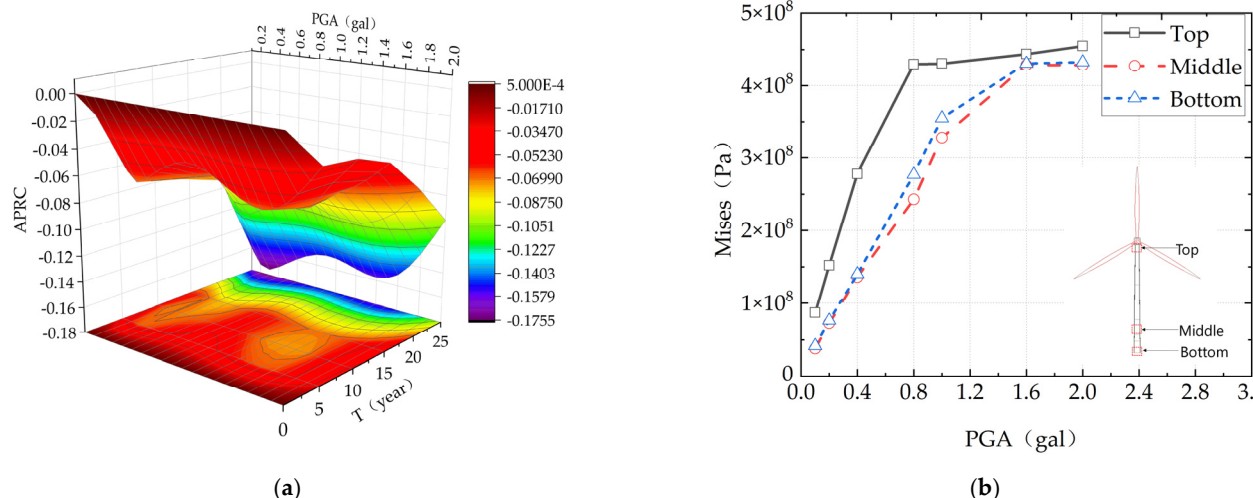

**Figure 14.** The Nihonkai-Chubu earthquake: (**a**) structural APRC value; (**b**) maximum Mises stress.

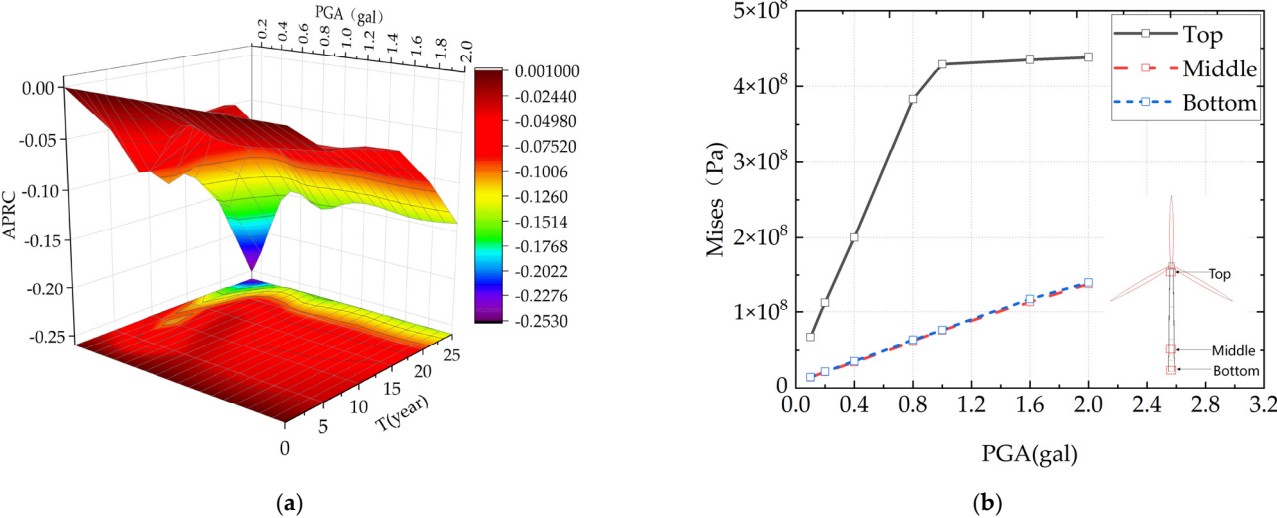

**Figure 15.** The Hyogoken-Nanbu earthquake: (**a**) structural APRC value; (**b**) maximum Mises stress.

With the increase in seismic peak acceleration, not only will the structural acceleration response be directly increased, but also the structural plasticity will be affected. When the seismic peak acceleration increases to change the structural elastic-plastic change, the structural acceleration response will change nonlinearly. In addition, under the same seismic acceleration, the long-term periodic earthquake in the far field and the direct downward earthquake in the near field bring the structural stress level to a higher level, and the material plasticity develops faster. The changes of structural APRC under three kinds of earthquakes show that the deeper corrosion degree increases the dynamic sensitivity of the structure.

### 3.4. Bending Moment-Curvature and Plasticity Rate Analysis of Wind Turbine Tower

Figure 16 shows the equivalent plastic strain values of the materials in the corrosion area of the wind power tower in 0, 10, 20 and 30 years when PGA = 2G under the Nihonkai-Chubu earthquake loading. It can be seen from Figure 16 that the maximum plastic strain value of the noncorrosive structure is 0.0038. With the increase of corrosion time, the plastic strain value of the material gradually increases, and the maximum plastic strain value of the structure corroded for 30 years reaches 0.015, which is 2.95 times larger than that of the noncorrosive structure. Moreover, it can be seen from the area surrounded by the plastic

strain curve that the corrosion leads to the acceleration of the plastic development of the tower in the corrosion area.

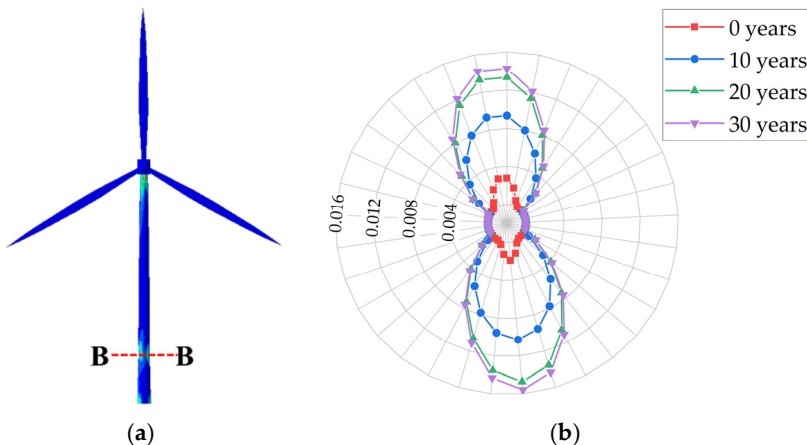

**Figure 16.** Equivalent plastic strain of corrosion area structure: (**a**) Section position; (**b**) Equivalent plastic strain of section B-B.

The yield curvature, maximum removal rate and structural plasticity rate of the tower structure under different corrosion years are calculated according to the guidelines for design of wind turbine support structures and foundations [35], as shown in Figure 17. It shows that the yield curvature of the tower structure does not change with the increase in corrosion years, but the maximum curvature decreases from 0.00217 m$^{-1}$ to 0.00184 m$^{-1}$. The corresponding structural plasticity rate is reduced from 2.306 to 1.954, and the seismic performance of the structure is obviously reduced.

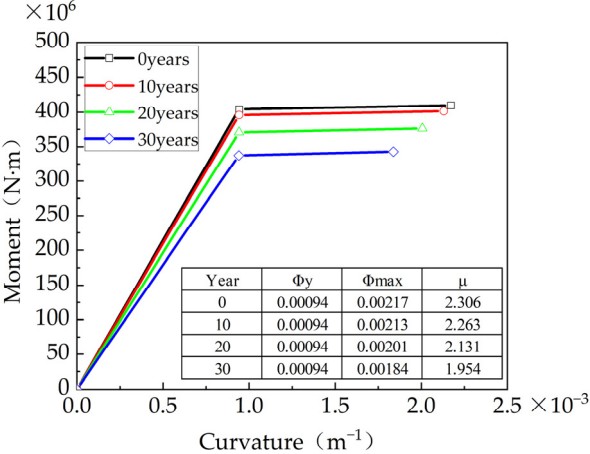

| Year | $\Phi y$ | $\Phi max$ | $\mu$ |
|---|---|---|---|
| 0 | 0.00094 | 0.00217 | 2.306 |
| 10 | 0.00094 | 0.00213 | 2.263 |
| 20 | 0.00094 | 0.00201 | 2.131 |
| 30 | 0.00094 | 0.00184 | 1.954 |

**Figure 17.** Bending moment curvature relationship of different corrosion structures.

In order to explore the influence of corrosion degree on the nonlinear development of wind turbine tower structures, the Nihonkai-Chubu earthquake of PGA = 2.0 g is selected. The moment curvature hysteretic curves of the wind turbine tower structure under four different corrosion conditions are calculated, respectively. The moment curvature hysteretic curve of the corrosion area is extracted, as shown in Figure 18. It shows that the maximum curvature of the noncorrosive structure under the Nihonkai-Chubu earthquake is 0.0013 m$^{-1}$, and the maximum plasticity rate is 1.38, which is less than the allowable plasticity rate of 2.306. The maximum curvature value of the structure corroded for 10 years is 0.003 m$^{-1}$, the maximum plastic rate is 3.14, which is larger than 2.263 higher than the maximum allowable plastic rate of the structure. The maximum curvature of the structure corroded for 20 years and 30 years is 0.0059 m$^{-1}$ and 0.0058 m$^{-1}$, respectively, and the

maximum plasticity rate is 6.27 and 6.16, respectively, which is beyond the maximum allowable plasticity rate of the structure. It can be seen from the fullness of the hysteresis loop that with the increase of corrosion time, the larger the area surrounded by the hysteresis loop, that is, when the structure is corroded, greater plastic deformation is needed to dissipate seismic energy under the same seismic action.

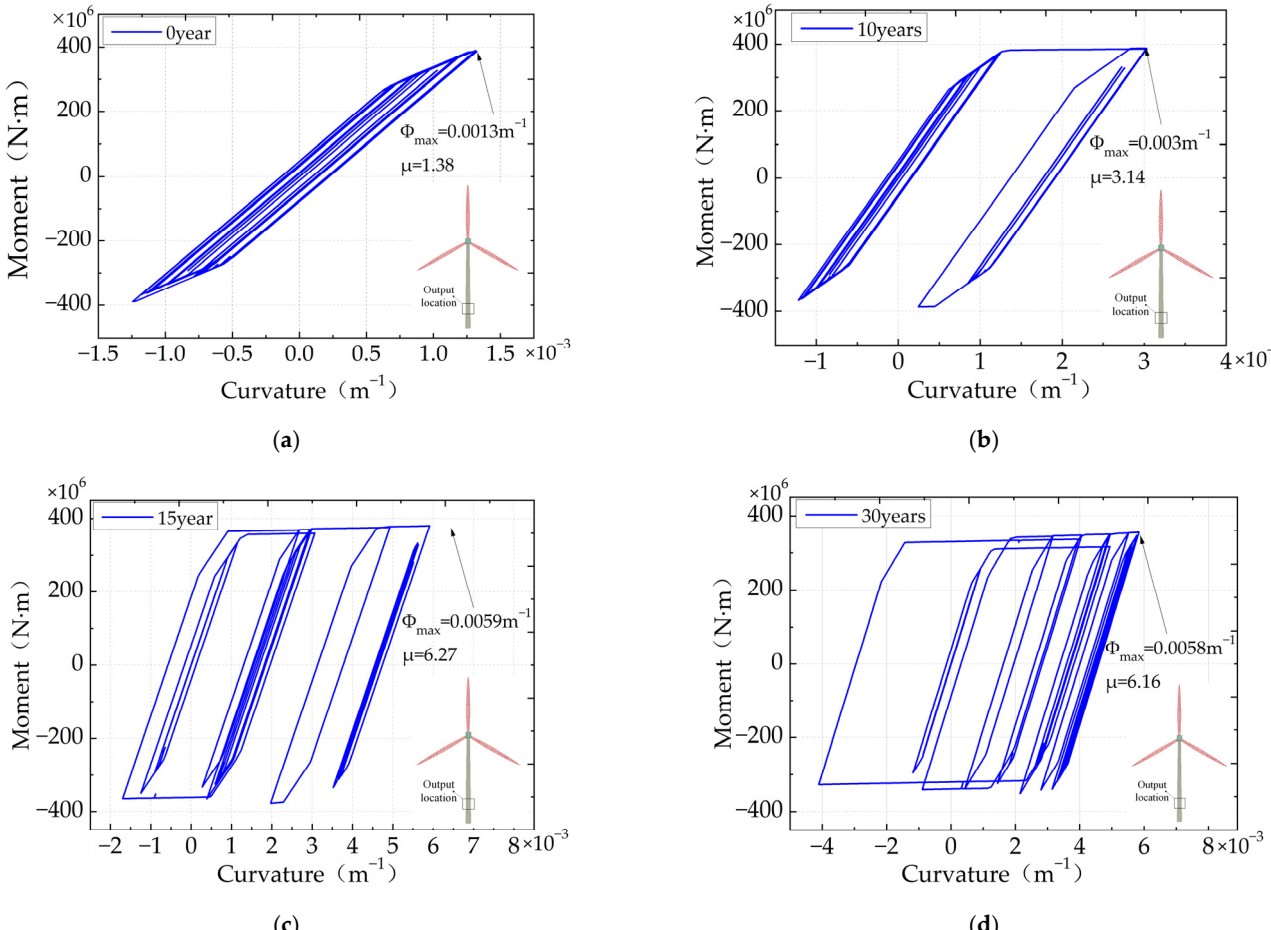

**Figure 18.** Moment Curvature hysteretic curve of structure under earthquake action: (**a**) Without corrosion; (**b**) 10 years of corrosion; (**c**) 20 years of corrosion; (**d**) 30 years of corrosion.

### 3.5. Affect of CFRP Reinforcement on Seismic Performance of Wind Turbine Tower

Take the wind power tower that has been corroded for 30 years as an example. The corrosion section of the wind power tower is reinforced with CFRP as shown in Figure 19. The CFRP model is created using the S4R unit in the ABAQUS2020 software. Use Create Composite Layup to set the number of fiber layers, the thickness of each fiber layer, and the ply angle of each fiber layer. The material properties of CFRP are expressed by the "ENGINEERING CONSTANTS" model and the "HASHIN DAMAGE" model. The CFRP material parameters are shown in Table 6.

The reinforcement height is 12 m. The number of CFRP layers is divided into two layers, four layers and eight layers. The CFRP laying direction is 0° and 90°, alternately. In order to make the structural damage more obvious, the Nihonkai-Chubu earthquake, with a smaller predominant frequency, was selected for loading. In order to fully develop the plasticity of the structure under earthquake action, the seismic peak acceleration is adjusted to 2G during loading.

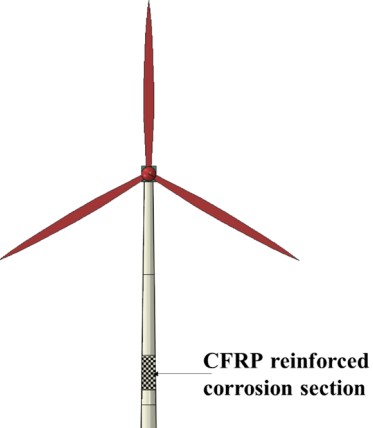

**Figure 19.** Wind turbine tower strengthened with CFRP.

**Table 6.** CFRP material parameters.

| Parameters | Values | Parameters | Values |
|---|---|---|---|
| $E_1$/GPa | 235 | $X_T$/MPa | 3400 |
| $E_2$/GPa | 17 | $X_C$/MPa | 1700 |
| $v_{12}$ | 0.34 | $Y_T$/MPa | 62 |
| $G_{12}$/GPa | 48 | $Y_C$/MPa | 190 |
| $G_{13}$/GPa | 48 | $S_{XY}$/MPa | 81 |
| $G_{23}$/GPa | 45 | $S_{YZ}$/MPa | 81 |

The displacement time history curve of the tower top and the residual deformation curve of the structure after the earthquake are shown in Figure 20. The influence of CFRP reinforcement on the displacement response of the structure on the second and fourth layers is similar. Compared with the reinforced structure, the displacement extreme value is reduced from 4.013 m to 3.687 m and 3.688 m, and the residual deformation of the tower top is reduced from 0.091 m to 0.067 m and 0.068 m. The reinforcement effect of 8-storey CFRP is more obvious than that of two-layer and four-layer reinforcement. The maximum displacement of the structure is reduced to 3.527 m, a decrease of 12.1%, and the residual displacement of the top is reduced to 0.014, a decrease of 8.5%. From the residual deformation of the structure along the tower height after the earthquake in Figure 20d, it can be seen that the residual deformation in the CFRP reinforced area is significantly reduced, and the effect of the eight layers of CFRP reinforcement is the most significant, which can reduce the structural corrosion area and the deformation of the tower top.

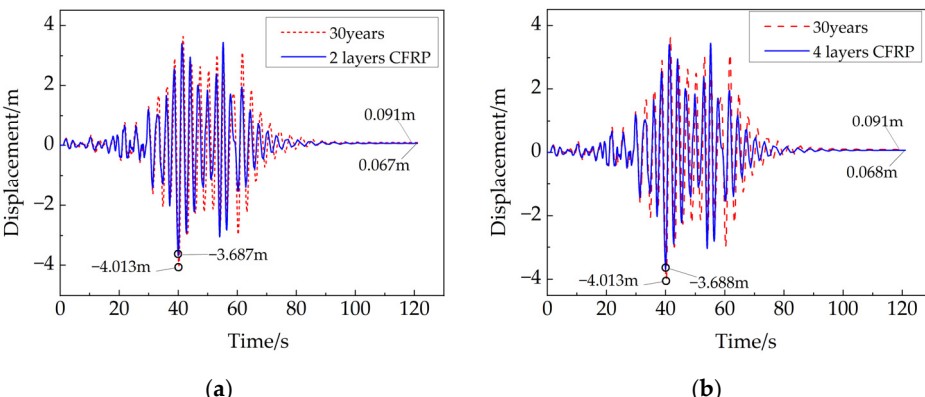

(**a**)        (**b**)

**Figure 20.** *Cont.*

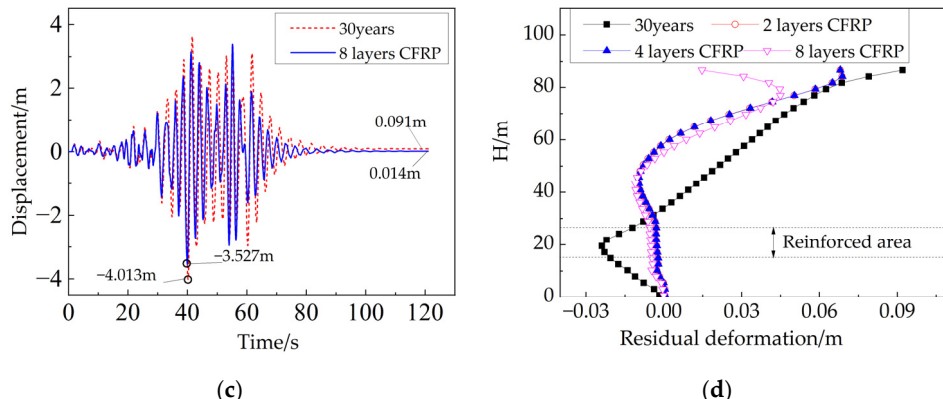

(c)                              (d)

**Figure 20.** Comparison of different reinforcement conditions: (**a**) two-layer CFRP reinforcement; (**b**) four-layer CFRP reinforcement; (**c**) eight-layer CFRP reinforcement; (**d**) Comparison of structural residual deformation.

It can be seen from Figure 21 that the main damage position of the structure that only strengthens the corrosion area is the upper tower, especially the top of the tower. This can also be obtained from the residual deformation of the structure. With the increase of CFRP layers, the damage degree of the steel pipe pile at the bottom of a wind power tower tends to increase.

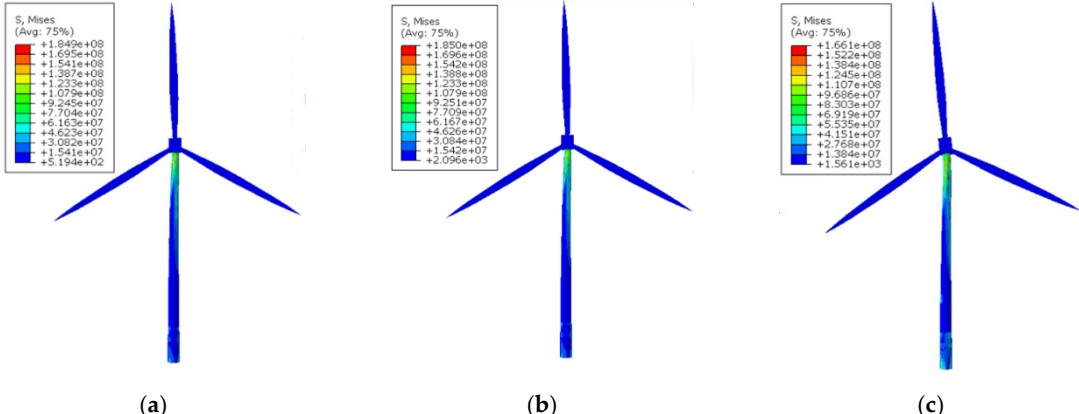

(a)                              (b)                              (c)

**Figure 21.** Structural stress cloud diagram: (**a**) two layer CFRP; (**b**) four layer CFRP; (**c**) eight layer CFRP.

## 4. Shaking Table Test

### 4.1. Shaking Table Test Design

On the basis of numerical analysis, the middle and lower sections of the structure are selected for the shaking table test. The shaking table model is shown in Figure 22. The influence of corrosion and CFRP reinforcement on the dynamic response of the structure is investigated.

The scale models of wind turbine towers with corrosion of 10, 20 and 30 years were designed and the shaking table experiment was carried out. Considering the limitations of experimental equipment size and other conditions, the similarity ratio of the model is 1/17 according to the research of Ian Prowell [36] and Ren Qianqian et al. [11]. The similarity relationship of each parameter is shown in Table 7.

The thickness of the tower in the corrosion area decreases according to the similarity ratio. The original section thickness of the non-corroded area remains unchanged. The bottom of the model is the simulation foundation embedded in the sand box. The towers are connected by flanges and bolts. The upper end of the tower barrel is provided

with a counterweight and connected to the tower body with bolts. Figure 23 shows the experimental model installation process.

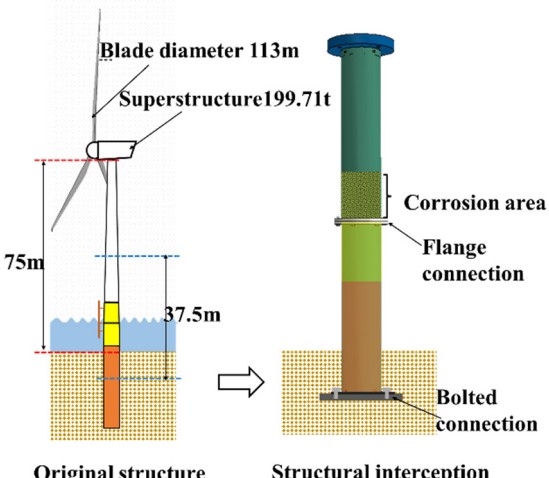

**Figure 22.** Selection of shaking table test model.

**Table 7.** Similarity ratio of experimental model.

| Parameter | Prototype | Model | Similarity Ratio (m/p) |
| --- | --- | --- | --- |
| $\sigma_y$ | 345 MPa | 345 MPa | 1 |
| $\varepsilon$ | $1.64 \times 10^{-3}$ m | $1.64 \times 10^{-3}$ m | 1 |
| E | $2.06 \times 10^{11}$ Pa | $2.06 \times 10^{11}$ Pa | 1 |
| $\sigma_u$ | 490 MPa | 490 MPa | 1 |
| $v$ | 0.3 | 0.3 | 1 |
| l | 42,500 mm | 2500 mm | 1/17 |
| D | 4500 mm | 270 mm | 1/17 |
| A | 699,004 mm$^2$ | 2418 mm$^2$ | 1/289 |
| P | 2,228,547.5 N | 7711 N | 1/289 |

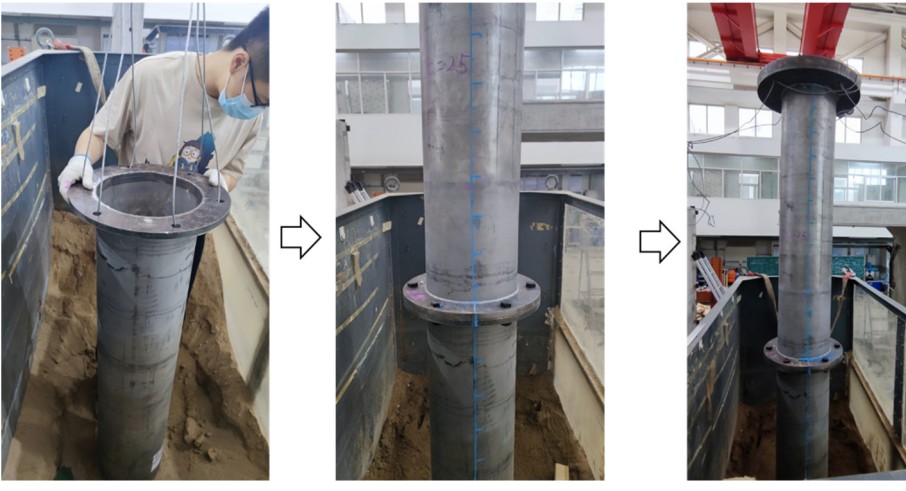

**Figure 23.** Installation of experimental model.

Figure 24 shows the layout of the acceleration sensor and the displacement sensor. The sensors are arranged along the loading direction, and a group is arranged every 300 mm from top to bottom.

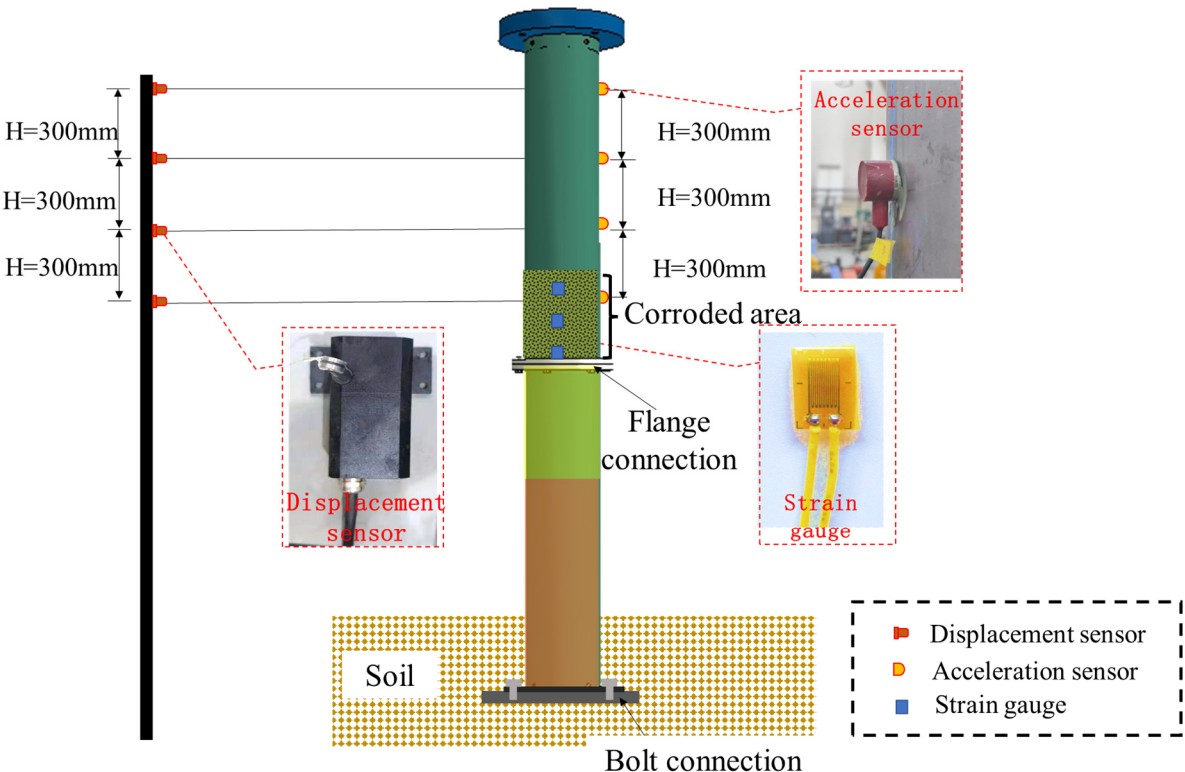

**Figure 24.** Sensor placement.

El Centro and Nihonkai-Chubu seismic waves are selected as seismic loads. In order to ensure the safety of the experiment, the peak value of the seismic wave is adjusted to 0.04 g and 0.1 g, respectively. The hydraulic vibration table and control equipment for loading are shown in Figure 25.

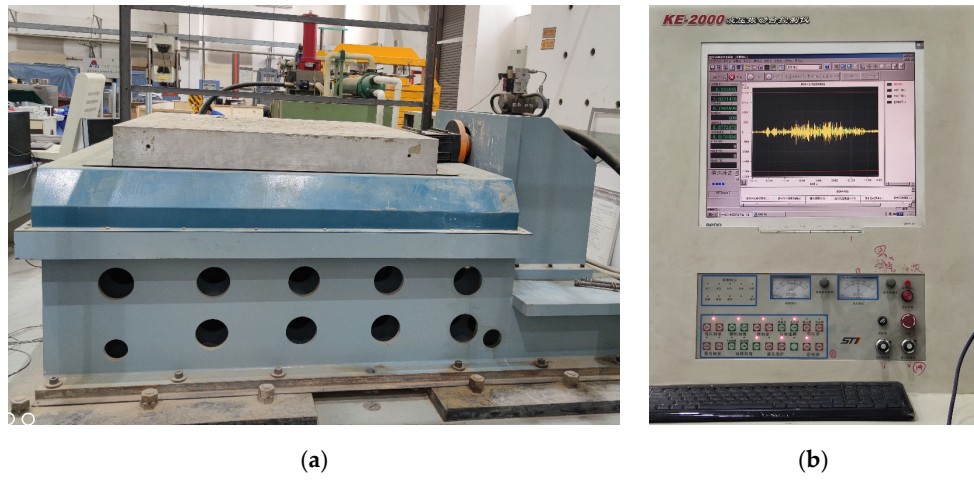

(**a**)                                                                                     (**b**)

**Figure 25.** Hydraulic vibration table control equipment: (**a**) Hydraulic vibration table; (**b**) Control equipment.

The loading conditions of the shaking table test are shown in Table 8. No. 1–No. 24 are the structural loads under different corrosion conditions without CFRP reinforcement, and No. 25~No. 42 are the loading profiles of corroded structures after CFRP reinforcement.

**Table 8.** Loading condition of shaking table test.

| Loading Condition | Experimental Condition | Wall Thickness/mm | Earthquake Records | PGA/Gal |
|---|---|---|---|---|
| No. 1 | El-01 | 3 | El Centro | 35 |
| No. 2 | El-02 | 3 | El Centro | 70 |
| No. 3 | Ni-01 | 3 | Nihonkai-Chubu | 35 |
| No. 4 | Ni-02 | 3 | Nihonkai-Chubu | 70 |
| No. 5 | Hy-01 | 3 | Hyogoken-Nanbu | 35 |
| No. 6 | Hy-02 | 3 | Hyogoken-Nanbu | 70 |
| No. 7 | El-03 | 2.8 | El Centro | 35 |
| No. 8 | El-04 | 2.8 | El Centro | 70 |
| No. 9 | Ni-03 | 2.8 | Nihonkai-Chubu | 35 |
| No. 10 | Ni-04 | 2.8 | Nihonkai-Chubu | 70 |
| No. 11 | Hy-03 | 2.8 | Hyogoken-Nanbu | 35 |
| No. 12 | Hy-04 | 2.8 | Hyogoken-Nanbu | 70 |
| No. 13 | El-05 | 2.65 | El Centro | 35 |
| No. 14 | El-06 | 2.65 | El Centro | 70 |
| No. 15 | Ni-05 | 2.65 | Nihonkai-Chubu | 35 |
| No. 16 | Ni-06 | 2.65 | Nihonkai-Chubu | 70 |
| No. 17 | Hy-05 | 2.65 | Hyogoken-Nanbu | 35 |
| No. 18 | Hy-06 | 2.65 | Hyogoken-Nanbu | 70 |
| No. 19 | El-07 | 2.5 | El Centro | 35 |
| No. 20 | El-08 | 2.5 | El Centro | 70 |
| No. 21 | Ni-07 | 2.5 | Nihonkai-Chubu | 35 |
| No. 22 | Ni-08 | 2.5 | Nihonkai-Chubu | 70 |
| No. 23 | Hy-07 | 2.5 | Hyogoken-Nanbu | 35 |
| No. 24 | Hy-08 | 2.5 | Hyogoken-Nanbu | 70 |
| No. 25 | CF-El-09 | 2.8 | El Centro | 35 |
| No. 26 | CF-El-10 | 2.8 | El Centro | 70 |
| No. 27 | CF-Ni-09 | 2.8 | Nihonkai-Chubu | 35 |
| No. 28 | CF-Ni-10 | 2.8 | Nihonkai-Chubu | 70 |
| No. 29 | CF-Hy-09 | 2.8 | Hyogoken-Nanbu | 35 |
| No. 30 | CF-Hy-10 | 2.8 | Hyogoken-Nanbu | 70 |
| No. 31 | CF-El-11 | 2.65 | El Centro | 35 |
| No. 32 | CF-El-12 | 2.65 | El Centro | 70 |
| No. 33 | CF-Ni-11 | 2.65 | Nihonkai-Chubu | 35 |
| No. 34 | CF-Ni-12 | 2.65 | Nihonkai-Chubu | 70 |
| No. 35 | CF-Hy-11 | 2.65 | Hyogoken-Nanbu | 35 |
| No. 36 | CF-Hy-12 | 2.65 | Hyogoken-Nanbu | 70 |
| No. 37 | CF-El-13 | 2.5 | El Centro | 35 |
| No. 38 | CF-El-14 | 2.5 | El Centro | 70 |
| No. 39 | CF-Ni-13 | 2.5 | Nihonkai-Chubu | 35 |
| No. 40 | CF-Ni-14 | 2.5 | Nihonkai-Chubu | 70 |
| No. 41 | CF-Hy-13 | 2.5 | Hyogoken-Nanbu | 35 |
| No. 42 | CF-Hy-14 | 2.5 | Hyogoken-Nanbu | 70 |

### 4.2. Affect of Corrosion on Dynamic Response of Structure

The acceleration time history curve of the top of the structure under the El Centro earthquake is shown in Figure 26. It can be seen that with the reduction of the wall thickness of the steel tower, the acceleration response of the top of the structure shows a downward trend, and the acceleration response of the corroded structure is smaller than that of the noncorroded structure in the whole seismic loading process. It shows this characteristic under different peak accelerations and different types of seismic waves.

It can be seen from Figure 27 that the corrosion causes the displacement of the top of the structure to increase, and the negative displacement of the structure is greater than the positive displacement of the structure. According to the analysis of the acceleration change at the top of the structure, the corrosion of the tower structure will lead to the local reduction of the structural stiffness, which will reduce the acceleration response and increase the displacement response of the structure under the earthquake action.

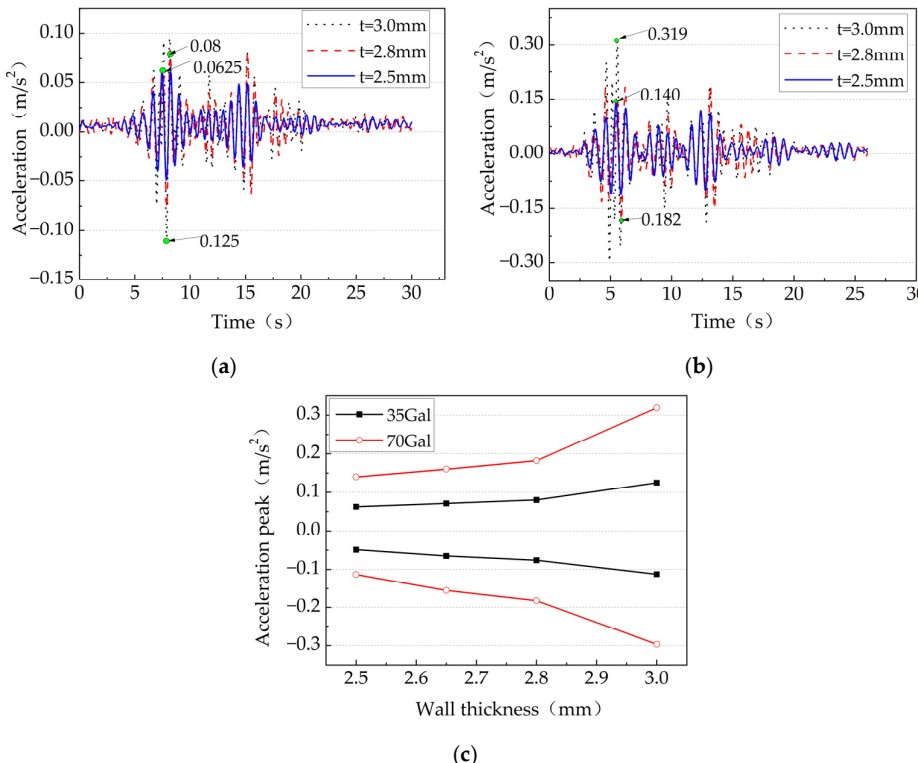

**Figure 26.** Comparison of structural accelerations under El Centro earthquake: (**a**) El Centro 35 Gal; (**b**) El Centro 70 Gal; (**c**) Comparison of acceleration peak.

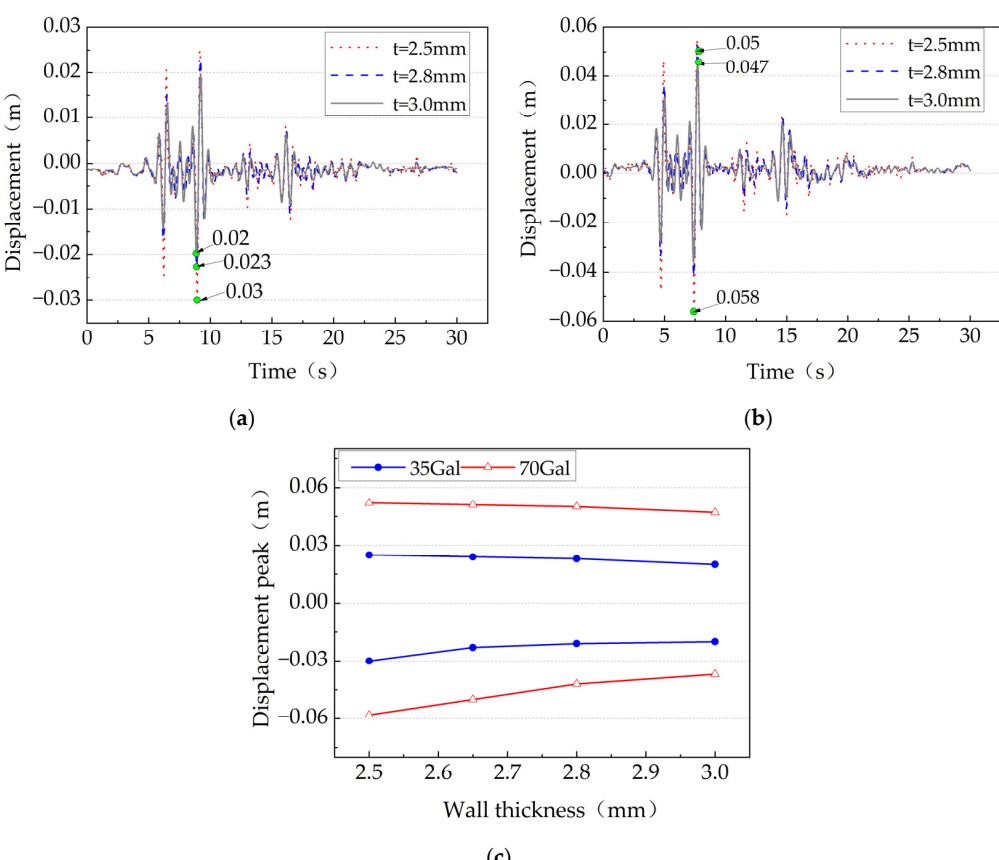

**Figure 27.** Displacement comparison under El Centro earthquake; (**a**) El Centro 35 Gal; (**b**) El Centro 70 Gal; (**c**) Comparison of displacement peak.

Figures 28 and 29 show the changes in peak acceleration and displacement at the top of the structure under the action of the Nihonkai-Chubu earthquake and the Hyogoken-Nanbu earthquake. The influence of corrosion on the seismic dynamic response of the wind power tower in the three kinds of earthquakes all shows the law that corrosion reduces the acceleration response of the tower and increases the displacement response of the tower. Since the steel is still in an elastic state under the action of 35 gal and 70 gal earthquakes, and the structure is not plastically damaged, the increase in the seismic peak does not make the displacement response of the structure significantly different. However, the peak acceleration of the structure has an enhancement effect when the seismic acceleration becomes larger, especially for the noncorrosive structure, the change in the structural response caused by the change in the seismic peak acceleration is greater. It can be inferred that the wind power tower structure with large rigidity is more sensitive to the change of seismic peak value, and the reduced rigidity of the corroded structure also makes the structure less sensitive to the change of seismic peak value.

### 4.3. Affect of CFRP Reinforcement on Dynamic Response of Structure

The dynamic response of the structure before and after CFRP reinforcement under different earthquake actions is compared. It can be seen from Figure 30 that the peak acceleration of the structure after CFRP reinforcement is increased, but there is still a large gap compared with the value of the noncorrosive structure. From the comparison of structural displacement extremum, it can be seen that the structural displacement extremum after CFRP reinforcement is significantly reduced, and the more severe the corrosion is, the more significant the reinforcement effect is. Combined with the hysteretic performance analysis of the CFRP steel composite structure, it can be seen that CFRP reinforcement contributes little to the improvement of the stiffness of the elastic section of the structure, so the reinforcement effect of CFRP is not significant when the structure is not damaged.

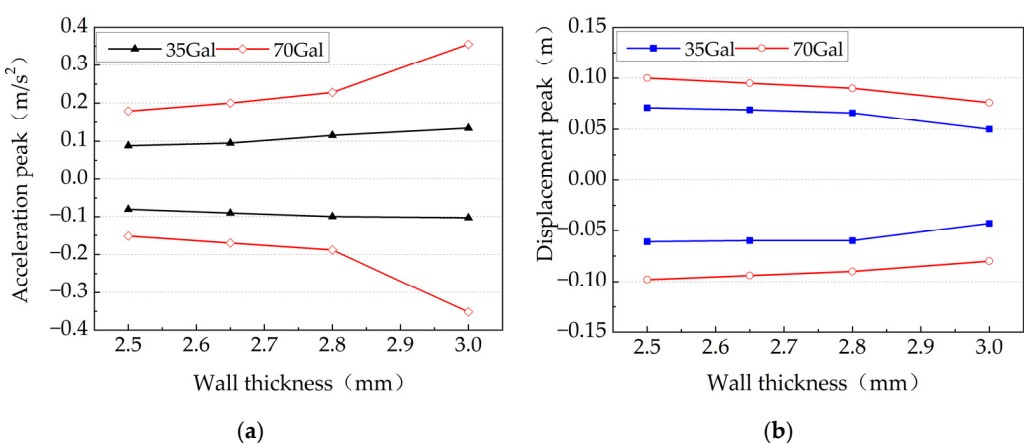

**Figure 28.** Dynamic response of structure under the Nihonkai-Chubu earthquake: (**a**) Comparison of acceleration peak; (**b**) Comparison of displacement peak.

Table 9 shows the comparison of structural acceleration response changes before and after structural reinforcement under different earthquake actions. It can be seen from the table that under the earthquake action of El Centro, the maximum acceleration peak of the structure strengthened by CFRP is 20.9%. Under the action of the Nihonkai-Chubu earthquake, the maximum increase in peak acceleration of the structure is 15.0%. Under the action of the Hyogoken-Nanbu earthquake, the maximum increase of the peak acceleration of the structure is 19.8%. Comparing the data of each group, it can be seen that the improvement of the CFRP reinforcement effect under far-field earthquakes is smaller than that of other types of earthquakes.

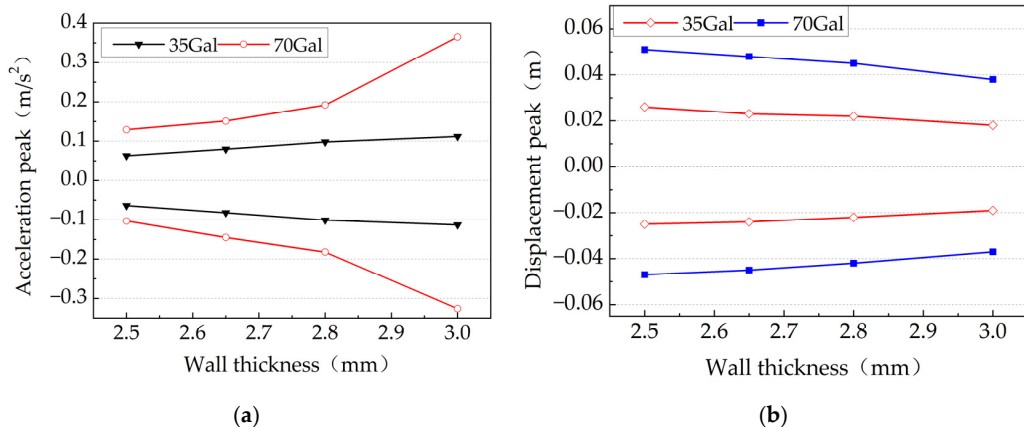

**Figure 29.** Dynamic response of structure under the Hyogoken-Nanbu earthquake: (**a**) Comparison of acceleration peak; (**b**) Comparison of displacement peak.

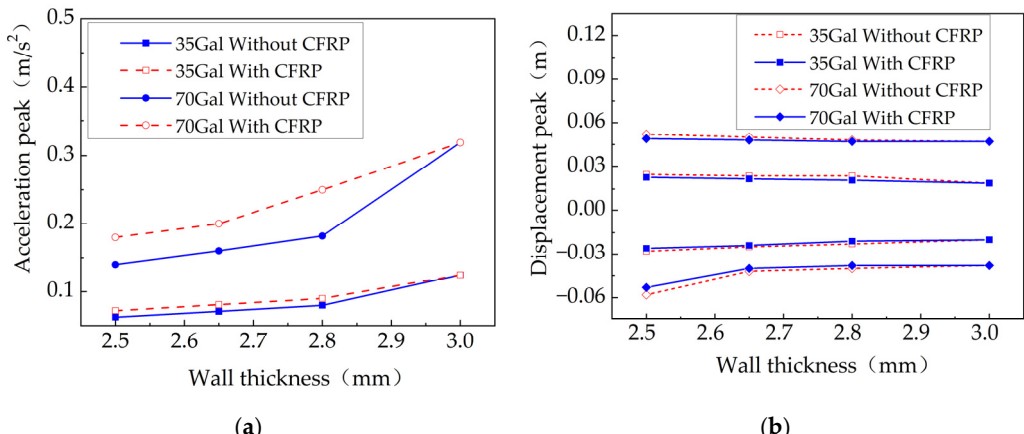

**Figure 30.** Comparison of structural dynamic response under El Centro earthquake: (**a**) Acceleration Peak comparison; (**b**) Displacement peak comparison.

**Table 9.** Comparison of structural acceleration response.

| Earthquake | Wall Thickness (mm) | 35 Gal | | Rate of Change | 70 Gal | | Rate of Change |
|---|---|---|---|---|---|---|---|
| | | Without CFRP | With CFRP | | Without CFRP | With CFRP | |
| El Centro | 2.5 | 0.0625 | 0.072 | 15.20% | 0.14 | 0.165 | 17.90% |
| | 2.65 | 0.071 | 0.081 | 14.10% | 0.16 | 0.18 | 12.50% |
| | 2.8 | 0.08 | 0.09 | 12.50% | 0.182 | 0.22 | 20.90% |
| | 3 | 0.125 | 0.125 | - | 0.319 | 0.319 | - |
| Nihonkai-Chubu | 2.5 | 0.088 | 0.093 | 5.70% | 0.179 | 0.19 | 6.10% |
| | 2.65 | 0.095 | 0.1 | 5.30% | 0.2 | 0.23 | 15.00% |
| | 2.8 | 0.115 | 0.12 | 4.30% | 0.228 | 0.26 | 14.00% |
| | 3 | 0.134 | 0.134 | - | 0.355 | 0.355 | - |
| Hyogoken-Nanbu | 2.5 | 0.063 | 0.075 | 19.00% | 0.129 | 0.15 | 16.30% |
| | 2.65 | 0.08 | 0.09 | 12.50% | 0.151 | 0.18 | 19.20% |
| | 2.8 | 0.098 | 0.106 | 8.20% | 0.192 | 0.23 | 19.80% |

### 4.4. Comparison of FE Simulation and Test Results

According to the shaking table test model, the finite element model of the wind turbine with the same size as the experiment is established using ABAQUS2020. The towers are

connected by flanges and bolts. The tower adopts the S4R element, and other components adopt the C3D8R element. The finite element model is shown in Figure 31.

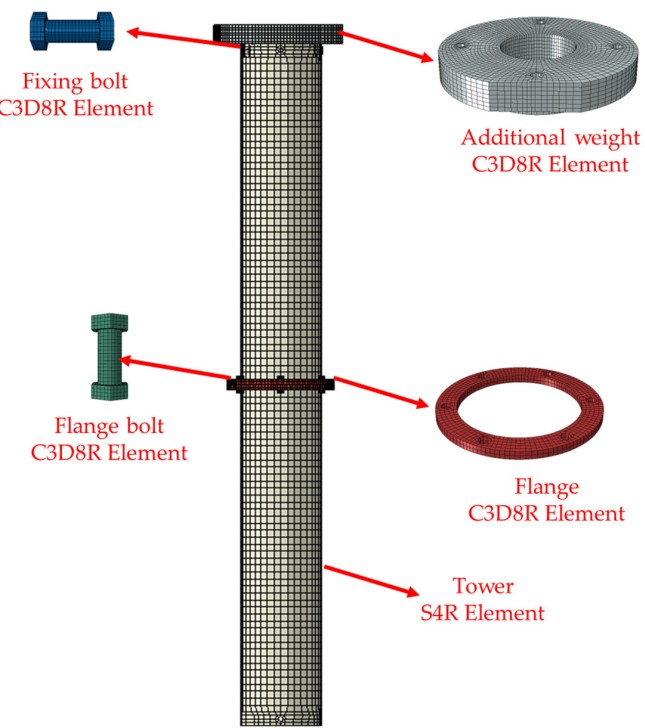

**Figure 31.** Scale finite element model.

The dynamic response characteristics of the structure under three different earthquake actions are simulated, respectively. The peak acceleration and displacement at the top of the tower are extracted and compared with the experimental results. As shown in Figures 32–34, where EX and FE represent experimental values and finite element simulation values, respectively.

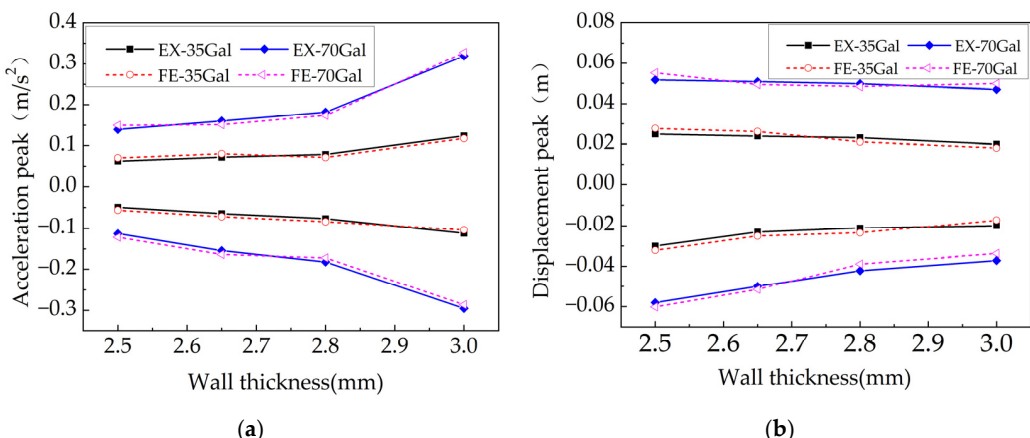

**Figure 32.** Comparison under the El Centro earthquake: (**a**) Comparison of acceleration peak; (**b**) Comparison of displacement peak.

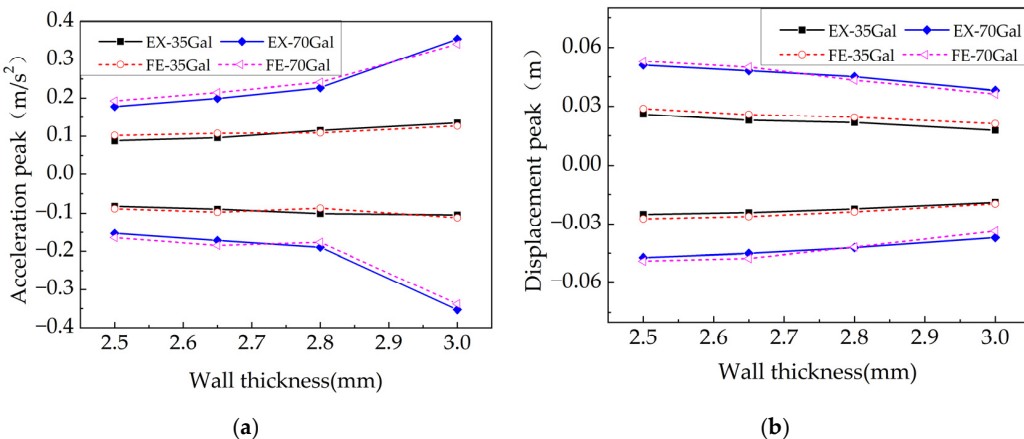

**Figure 33.** Comparison under the Nihonkai-Chubu earthquake: (**a**) Comparison of acceleration peak; (**b**) Comparison of displacement peak.

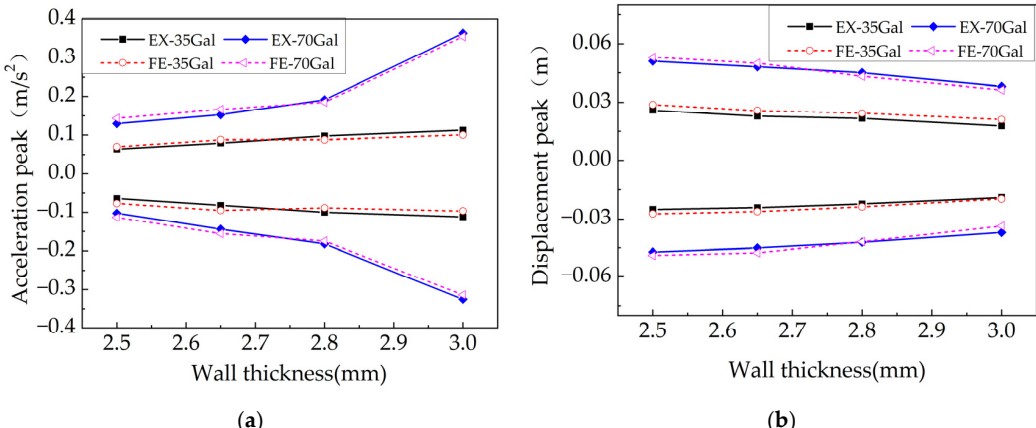

**Figure 34.** Comparison under the Hyogoken-Nanbu earthquake: (**a**) Comparison of acceleration peak; (**b**) Comparison of displacement peak.

Comparing the experimental results and finite element simulation results under three different earthquakes. It can be seen that the difference value of peak acceleration is 2.4~16.8%. Three of these data sets exceed 15%. Other difference ranges are within the allowable range of the project. The range of peak displacement differences is 0.6~16.7%. There are also three groups of data exceeding 15%. Other difference ranges are within the allowable range of the project. The proportion of numerical differences exceeding the limit is 6.25%. It can be concluded that the finite element simulation results have good relativity with the experimental results.

## 5. Conclusions

In this study, the influence of corrosion and different earthquake types on a 3 MW offshore wind turbine tower is studied by numerical simulation field test and shaking table experiment, and the corresponding influence of near-field earthquakes and far-field earthquakes on the structural dynamics of wind turbine towers is compared. At the same time, the incremental dynamic analysis method is used to study the influence of corrosion on the structural buckling and collapse of wind turbine towers. The research conclusions of this paper are universal for single-pile fans. Other types of single-pile fans can refer to the conclusions of this paper for reinforcement design.

1. The dynamic response of wind turbine towers is more sensitive to near-field earthquakes. Under the same PGA increment, the acceleration and displacement increment increase by 11% and 78%, respectively, compared with the far-field earthquake.

2.	The plastic ratio of the structure with serious corrosion is smaller than that without corrosion. Under the earthquake, the area surrounded by the moment curvature hysteretic curve of the splash area increases gradually.

3.	The shaking table test results show that the dynamic response of the structure under different corrosion conditions is significantly different, and the peak acceleration decreases with the increase in corrosion degree.

4.	Using CFRP to reinforce the corrosion area of the tower can effectively reduce the dynamic response of the wind turbine tower and improve the seismic performance of the structure.

5.	In order to better understand the corrosion condition of the wind turbine tower in service, it is recommended to regularly use a long-range laser Doppler vibrometer and other equipment to monitor the change of the natural frequency of the wind turbine or the thickness of the steel pipe pile. According to the monitoring data, a reasonable reinforcement scheme should be designated as early as possible.

**Author Contributions:** Conceptualization, B.S.; methodology, D.W.; software, D.W. and C.L.; validation S.D.; writing—original draft preparation, D.W. and S.D.; writing—review and editing, D.W. and C.W.; supervision, B.S. All authors have read and agreed to the published version of the manuscript.

**Funding:** This research is supported by the National Natural Science Foundation of China (52078038).

**Institutional Review Board Statement:** Not applicable.

**Informed Consent Statement:** Not applicable.

**Data Availability Statement:** Some or all the data and models that support the findings of this study are available from the corresponding author upon reasonable request.

**Acknowledgments:** Thanks to Jiangsu Rudong Electric Power Co., Ltd. for its help in this study.

**Conflicts of Interest:** The authors declare no conflict of interest.

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
