# Peer review of "Seismic Effect of Marine Corrosion and CFRP Reinforcement on Wind Turbine Tower"

_applsci, doi:10.3390/app121910136_

Round 1
Reviewer 1 Report
The Paper is focused on the marine corrosion effect on the dynamic response of offshore turbine tower. CFRP material have been proposed to reinforce the tower. Numerical results were perfomed validated by scaled shake table tests. The paper is well written and really interesting. The topic is worth of investigation. Some comments are:
1 - why wind load was not considered ?
2 - From figure 3 seems that the tower of the wind turbine is made by more than one steel pieces (presence of a flange connection). On the contrary the finite element model is a unique piece. Since the flange connection is one of the thing that inflence the tower response it is necessary to justify better the model assumptions and to describe more in detail the real tower.
3- the dynamic validation of the model is based on figure 8, but is not completely clear to me. In the graph seems to be a very low damping associated to the frequency indicated. Please justify this choice.
4 - for the dynamic validation model of wind turbine make reference to chapter 2 of the following paper https://doi.org/10.1016/j.istruc.2021.02.053
5 - Despite the experimental results are interesting it is hard to assume the validity of this studio and the relathionship between this results and the one from the numerical model. Did you create a model also of the scaled tests ?
6 - Some typos were detected along the text
7 - In the conclusion you should add a suggestion on how detect the corrosion state of a real tower by using remote sensors-continuosly monitoring system.
Author Response
First of all, thank you for your valuable comments on this manuscript, which has greatly benefited me. According to your comments, I have revised the manuscript and marked it in the manuscript. In addition, if you think the manuscript or research still needs to be improved, please do not hesitate to comment. Please see the attachment for the specific reply to the comments.Sincerely thank you for your guidance and help.

Reviewer 2 Report
main storyline is clear and the work load is enough to support your final conclusion.
A few comments here
1) when comes to abbreviation, please list their full expressions first, like what is CFRP and FE
2) define every symbol you used appropriately, like in equation 1, what is T
3) for publication quality, please be careful about the details, like figure 30 why the legend goes out of the plot box, how to display the figures if the number of subfigures are three. Also for the equation listed
4) on top of the comments aforementioned, please argue why you choose one specific method at where is needed, like all test on Hua Rui wind turbine tower, the same conclusion applies for other type of wind turbine tower? why embedded point method and equation 1 as well as 2 is good to model the system? the corrosion probability mode of Weibull function why is OK for marine corrosion? How to model CFRP in your model?
Author Response

(The authors gave the same response as above.)

Round 2
Reviewer 2 Report
Now the manuscript is more clear.